# The landscape of coadaptation in *Vibrio parahaemolyticus*

**Yujun Cui**[1†]*, **Chao Yang**[1,2†], **Hongling Qiu**[3,4†], **Hui Wang**[3,4], **Ruifu Yang**[1], **Daniel Falush**[5]*

[1]State Key Laboratory of Pathogen and Biosecurity, Beijing Institute of Microbiology and Epidemiology, Beijing, China; [2]Shenzhen Centre for Disease Control and Prevention, Shenzhen, China; [3]School of Public Health, Shanghai Jiao Tong University School of Medicine, Shanghai, China; [4]Institute for Nutritional Sciences, Chinese Academy of Sciences, Shanghai, China; [5]The Center for Microbes, Development and Health, Key Laboratory of Molecular Virology and Immunology, Institut Pasteur of Shanghai, Chinese Academy of Sciences, Shanghai, China

**Abstract** Investigating fitness interactions in natural populations remains a considerable challenge. We take advantage of the unique population structure of *Vibrio parahaemolyticus*, a bacterial pathogen of humans and shrimp, to perform a genome-wide screen for coadapted genetic elements. We identified 90 interaction groups (IGs) involving 1,560 coding genes. 82 IGs are between accessory genes, many of which have functions related to carbohydrate transport and metabolism. Only 8 involve both core and accessory genomes. The largest includes 1,540 SNPs in 82 genes and 338 accessory genome elements, many involved in lateral flagella and cell wall biogenesis. The interactions have a complex hierarchical structure encoding at least four distinct ecological strategies. One strategy involves a divergent profile in multiple genome regions, while the others involve fewer genes and are more plastic. Our results imply that most genetic alliances are ephemeral but that increasingly complex strategies can evolve and eventually cause speciation.

*For correspondence:
cuiyujun.new@gmail.com (YC);
danielfalush@googlemail.com (DF)

†These authors contributed equally to this work

**Competing interests:** The authors declare that no competing interests exist.

## Introduction

The importance of coadaptation to evolution was recognized by Darwin in the 6[th] edition of Origin of Species, where he wrote: 'In order that an animal should acquire some structure specially and largely developed, it is almost indispensable that several other parts should be modified and coadapted' (*Darwin, 1859*). As Darwin's argument implies, complex phenotypic innovation require adaptation at multiple genes and it is inevitable that some of the changes involved will be costly on the original genetic background, implying epistasis - that is non-additive fitness interactions - between adaptive loci.

The consequences of epistasis for the evolution of phenotypic diversity depends on transmission genetics, that is how genetic material is passed from one generation to the next, and population structure. For example, in outbreeding animals, mating mixes up variation every generation. A consequence of continuous reassortment is that genes only increase in frequency if they have high average fitness across genetic backgrounds (*Neher and Shraiman, 2011*). As a result, extensive linkage disequilibrium (LD) due to natural selection is rare (*Pritchard and Przeworski, 2001*) and it is difficult to maintain dissimilar genetic strategies concurrently in the same population unless the strategies are encoded by a small number of loci. This means that the coadaptation necessary for extensive phenotypic diversification can only take place when facilitated by barriers to gene flow, such as geographical separation, mate choice or the suppression of recombination for example by inversion polymorphisms (*Dobzhansky, 1937*; *Wallace, 1991*). This feature also makes it difficult to

study the process of complex coadaptation without temporally sampled genetic data, which remains rare despite the advent of technology for sequencing ancient DNA.

In bacterial populations, recombination happens much less frequently than cell division. For example, *Vibrio parahaemolyticus* lives in coastal waters and causes gastroenteritis in humans and economically devastating diseases in farmed shrimps. It is capable of replicating in less than 10 min in appropriate conditions (*Makino et al., 2003*), while the doubling time in the wild of the related bacteria *Vibrio cholerae* has been estimated as slightly over an hour (*Gibson et al., 1880*). Approximately 0.017% of the genome recombines each year (*Yang et al., 2019a*), implying that there are approximately 50 million generations on average between recombination events at a given genetic locus. The consequence of these transmission genetics are that mutations that are beneficial only on specific backgrounds have a chance to rise to high frequency on those backgrounds even if they are harmful on others. Epistatic interactions that involve only small selective coefficients *s* (for example $s = 1.0 \times 10^{-4}$) can therefore create an imprint on the genome in the form of strong LD (*Arnold et al., 2018*).

Although recombination happens slowly on the timescale of bacterial generations, the Asian population of *V. parahaemolyticus* has had a large effective population size for at least the last 15,000 years, or approximately 130 million bacterial generations (*Yang et al., 2018*). As a result, the population of *V. parahaemolyticus* is unusual amongst bacteria in that there is approximate linkage equilibrium between most loci greater than 3 kb apart on the chromosome (*Yang et al., 2018*). This feature increases the power of tests for interaction based on identifying non-random associations, which relies on identifying the same combination of alleles on independent genetic backgrounds and can therefore be confounded by clonal or population structure unless this is appropriately controlled for.

We perform a systematic screen for coadaptation in the core and accessory genome, based on a larger sample of genomes than (*Cui et al., 2015*), here for the first time performing a screen for the co-occurrence of accessory genome elements. We have taken a conservative approach to identifying statistical associations, rigorously filtering the set of genomes used in our discovery dataset to eliminate any hint of population structure. We performed more than 14 billion Fisher exact tests between variants in the core and accessory genomes, using a cut-off of P < 10⁻¹⁰ with the aim of assembling a comprehensive list of common genetic variants that have strong LD between them due to fitness interactions.

We find that the great majority of interactions involve small numbers of accessory genome elements, with surprisingly little involvement of the core genome. However, we also identify a complex multi-locus interaction, in which core and accessory genomes have evolved in parallel to create coadapted gene complexes encoding at least four distinct strategies. We show that it is possible to characterize these strategies, both at the level of strains and at the level of the genetic variants encoding them, using hierarchical clustering. Our results demonstrate that *V. parahaemolyticus* have progressively modified their own fitness landscape through coadaptation and demonstrate the fundamental importance of lateral flagella variation to their ecology.

## Results

### Detection and characterization of interaction groups

The presence of clonal and population structure within a dataset results in genome-wide LD that can confound screens for epistatic interactions based on LD patterns. To minimize the effect of population structure we first removed closely related strains within our global collection of 1,103 isolates (*Yang et al., 2019a*) to generate a non-redundant set of 469 genomes that each differed from each other by more than 2,000 SNPs (*Figure 1a*). As previously described, the sample is subdivided into four distinct populations, which have different, although currently overlapping, geographical distributions (*Yang et al., 2019a*). These four populations are clearly visible in Principle Coordination Analysis (PCoA) of the SNP variants in this dataset (*Figure 1b*). The great majority of strains in our sample are from the Asian population, VppAsia, so we excluded isolates from other populations. We then used fineSTRUCTURE (*Lawson et al., 2012*) to look for signals of clonal structure within the remaining isolates, iteratively removing isolates until all of the isolates were assigned to a single

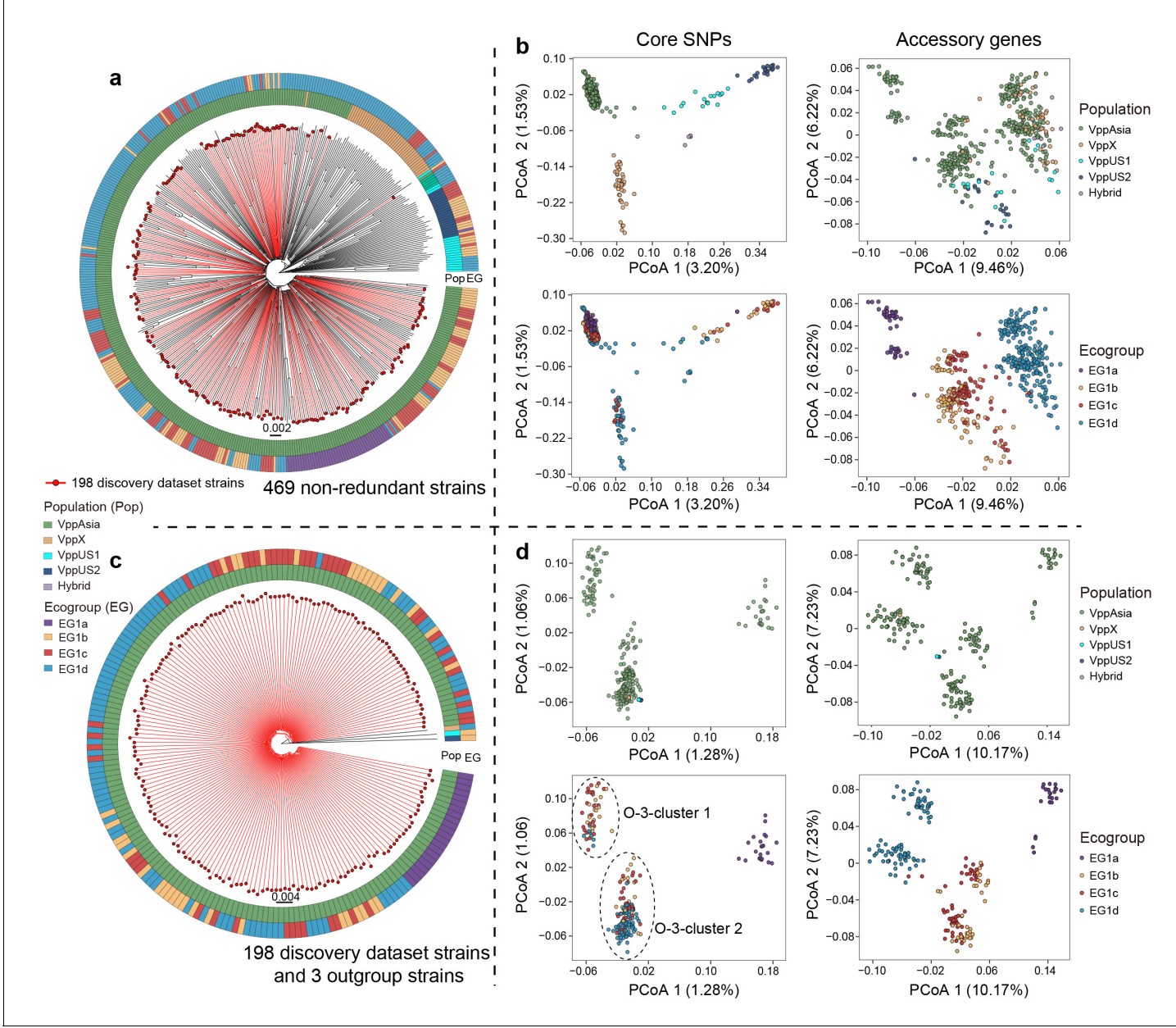

**Figure 1.** Neighbor-joining (NJ) trees and principle coordination analysis (PCoA) of 469 non-redundant dataset and 198 strain discovery dataset. (a, c) NJ trees of strains from two datasets. Red branches indicate strains in the discovery dataset. The colored circles indicate populations (inner) and ecogroups (outer) according to the legend on the left. (b, d) PCoA analysis of strains from two datasets based on core SNPs and accessory genes. Colors of points indicate the populations and ecogroups according to the legend on the right. Two distinct genotype clusters separated by O-3 variants were highlighted with dashed ellipses.

The online version of this article includes the following figure supplement(s) for figure 1:

**Figure supplement 1.** fineSTRUCTURE showing no clonal frame in the discovery dataset of 198 VppAsia isolates.

population (Materials and methods), leading to a discovery dataset of 198 strains (*Figure 1c*, *Figure 1—figure supplement 1*).

Using the discovery dataset, we performed a Fisher exact test of associations between all pairwise combinations of 151,957 SNP variants within the core and 14,486 accessory genome elements. As has been observed previously (*Cui et al., 2015*), most of strong associations occurred between sites within 3 kb on the chromosome (*Figure 2*). In order to exclude associations that arise only due to physical linkage, we excluded all sets of associations that spanned less than 3 kb, including

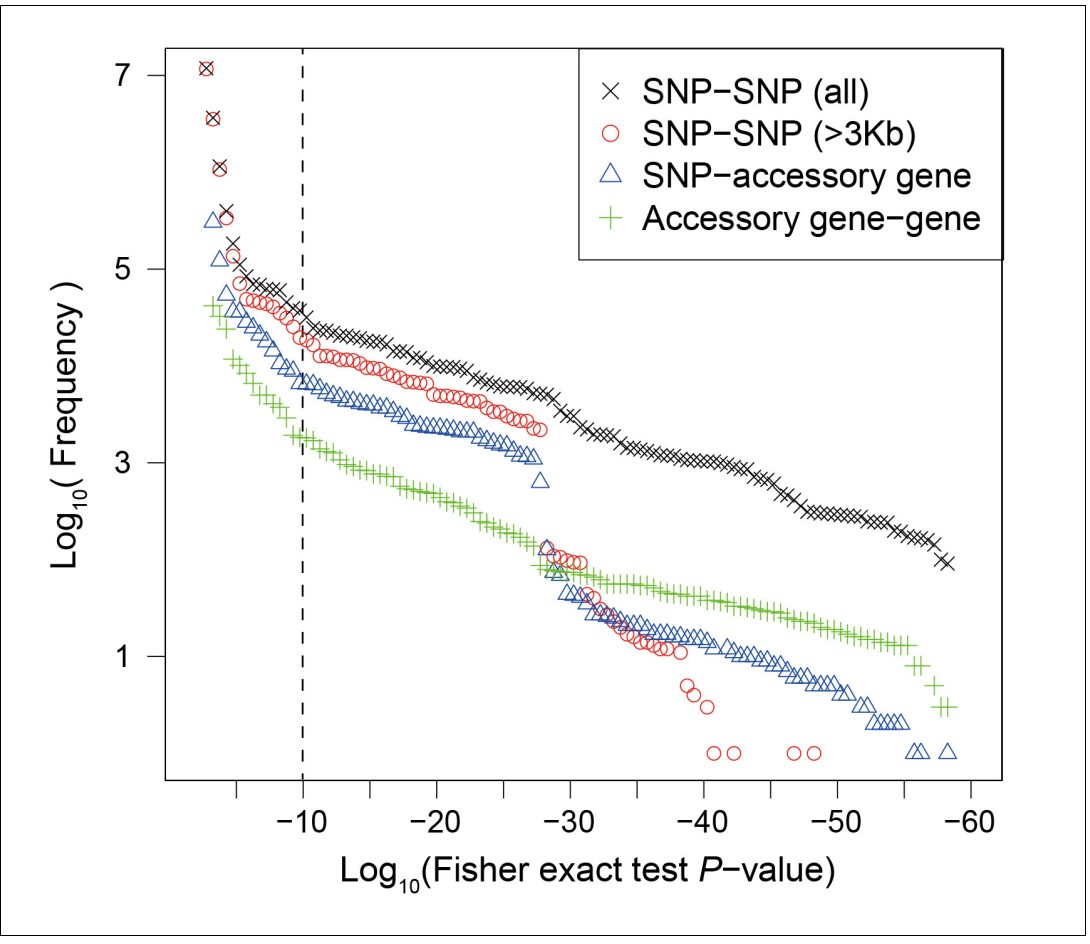

**Figure 2.** Frequency distribution of Fisher exact test *P* values between genetic variants. Colors and shapes indicate the interactions between different types of variants. The vertical dotted line shows the threshold p=$10^{-10}$.

between accessory genome elements. This left us with 452,849 interactions with P < $10^{-10}$, which grouped into 90 interaction groups (IG, a set of SNP/accessory gene connected by at least one significant interaction), all of which involved at least one accessory-genome element, with 8 also including core genome SNPs and 35 including multiple genome regions. In total these IGs included 1,873 SNPs in 100 core genes and 1,460 accessory genome elements (*Table 1*). Interacting SNPs were

**Table 1.** Summary of interactions detected in coadaptation screen.

| | Total number | IG1 | | IG2-90 | |
|---|---|---|---|---|---|
| | | Number | Fraction | Number | Fraction |
| SNP-SNP pair | $2.3 \times 10^{10}$ | 289186 | 0.00% | 22751 | 0.00% |
| SNP-Accessory gene pair | $2.2 \times 10^{9}$ | 113973 | 0.01% | 1188 | 0.00% |
| Accessory gene-gene pair | $2.1 \times 10^{8}$ | 13487 | 0.01% | 12264 | 0.01% |
| SNP | 151957 | 1540 | 1.01% | 333 | 0.22% |
| Synonymous (Syn) | 117541 | 1084 | 0.92% | 226 | 0.19% |
| Nonsynonymous (NonSyn) | 23673 | 379 | 1.60% | 107 | 0.45% |
| NonSyn/Syn | 0.2 | 0.35 | | 0.47 | |
| Core gene | 3936 | 82 | 2.25% | 18 | 0.56% |
| Accessory gene | 14486 | 338 | 2.33% | 1122 | 7.75% |

enriched for non-synonymous variants (26% vs. 16%, $P < 0.01$, Fisher exact test), which is consistent with natural selection being the force generating the linkage disequilibrium we detected.

## Complex structure of interaction group 1

The largest interaction group 1 (IG1) accounted for the majority of interacting SNPs (82%) as well as a significant fraction of accessory genome elements (23%), while IG2-IG90 generally consisted of a small number of interacting SNPs (0-297) and accessory genes (2-128). IG1 is shown in *Figure 3*, *Figure 3—figure supplements 1–3*, *Figure 4* and *Supplementary file 1*, while IG2-IG90 are displayed in *Figure 5*, *Figure 5—figure supplement 1*, *Figure 6* and *Supplementary file 2*. In order to interpret these associations in a broader ecological and geographical context, we performed further analyses on the interactions that we identified in the discovery dataset on the non-redundant set of 469 strains.

IG1 involves a large number of pairwise interactions, presenting a challenge for interpretation. We filtered the results to remove putatively-non-causal interactions using ARACNE (*Margolin et al., 2006*), but this only reduced the number of interactions from 414,785 to 103,241, which is still far too many to interpret (*Supplementary file 1*). However, hierarchical clustering revealed that strains fall into four distinct 'ecogroups' (EGs) based on IG1 variants (*Figure 3a,b*). Three of these groups, EG1a, EG1b and EG1d have a large number of variants (50-965) that are associated with them, and we used the clustering to sort these variants into tiers, with 'Tier 1' (T1) corresponding to the variants that showed the strongest association for each EG (*Figure 3*, *Table 2*, *Supplementary file 2*). Associated variants also show other patterns, for example, Tier"Other 1' (O-1) variants distinguish EG1a and EG1b from other EGs, with some exceptions; O-2 variants distinguish EG1a and EG1d from other EGs, also with exceptions; O-3 variants are polymorphic in EG1b and EG1c but mostly fixed in EG1a and EG1d (*Figure 3a,b*). Similar but not identical structuring of strains was obtained if clustering was performed only using core genome SNPs (*Figure 3—figure supplement 1a*) or accessory genome elements (*Figure 3—figure supplement 1b*).

Although fineSTRUCTURE did not detect any population structure within the discovery dataset, PCoA analysis of SNPs identifies axes of variation that reflect the ecogroup structure within EG1 (*Figure 1d*). PCo1, which explains 1.28% of the total SNP variance differentiates EG1a strains from the others while PCo2, which explains 1.06% of the variance differentiates the two distinct genotype clusters separated by O-3 variants (*Figure 1d*), with O-3-cluster 1 corresponding to SNP Allele 2 shown in *Figure 3a*. The phylogenetic analysis also reveals a split between EG1a and the remaining isolates (*Figure 1a, c*).

The reason that PCoA and phylogenetic analysis reveals population structure within the discovery dataset, while fineSTRUCTURE does not, is because the differentiated loci are sharply concentrated into a small number of chromosomal regions, while fineSTRUCTURE looks for evidence of higher sharing across the entire genome. Chromosome painting (*Figure 3d*) shows evidence for sharp peaks of differentiation of different EGs around the IG1 loci identified by our coadaptation scan, especially for EG1a strains, but with little evidence for differentiation elsewhere. This pattern is qualitatively distinct either for that observed between geographically differentiated populations or for clonally related strains, which show higher-than-expected copying probability (probability of genetic variants inherited from the same population or ecogroup) throughout the genome but without sharp peaks (*Figure 3—figure supplement 2*). These results are consistent with there being a common gene pool shared by all of the ecogroups within each *V. parahaemolyticus* population, with ecogroup structure being maintained by natural selection rather than barriers to gene flow.

The accessory genome elements are present in many of the same configurations as core genome SNPs, but with ecogroup structure explaining a much larger fraction of the overall variation. The first PCo accounts for around 10% of the variance, both for the discovery dataset and the non-redundant dataset, and nearly cleanly distinguishes EG1a strains from EG1b, c strains and from EG1d strains (*Figure 1b,d*). Geographically based population structure is correspondingly less important, with weak differentiation of two of the populations (VppAsia and VppX) from the other two evident within PCo2 (*Figure 1b,d*), which is consistent with our previous observation that few population-specific genes were identified (*Yang et al., 2019a*).

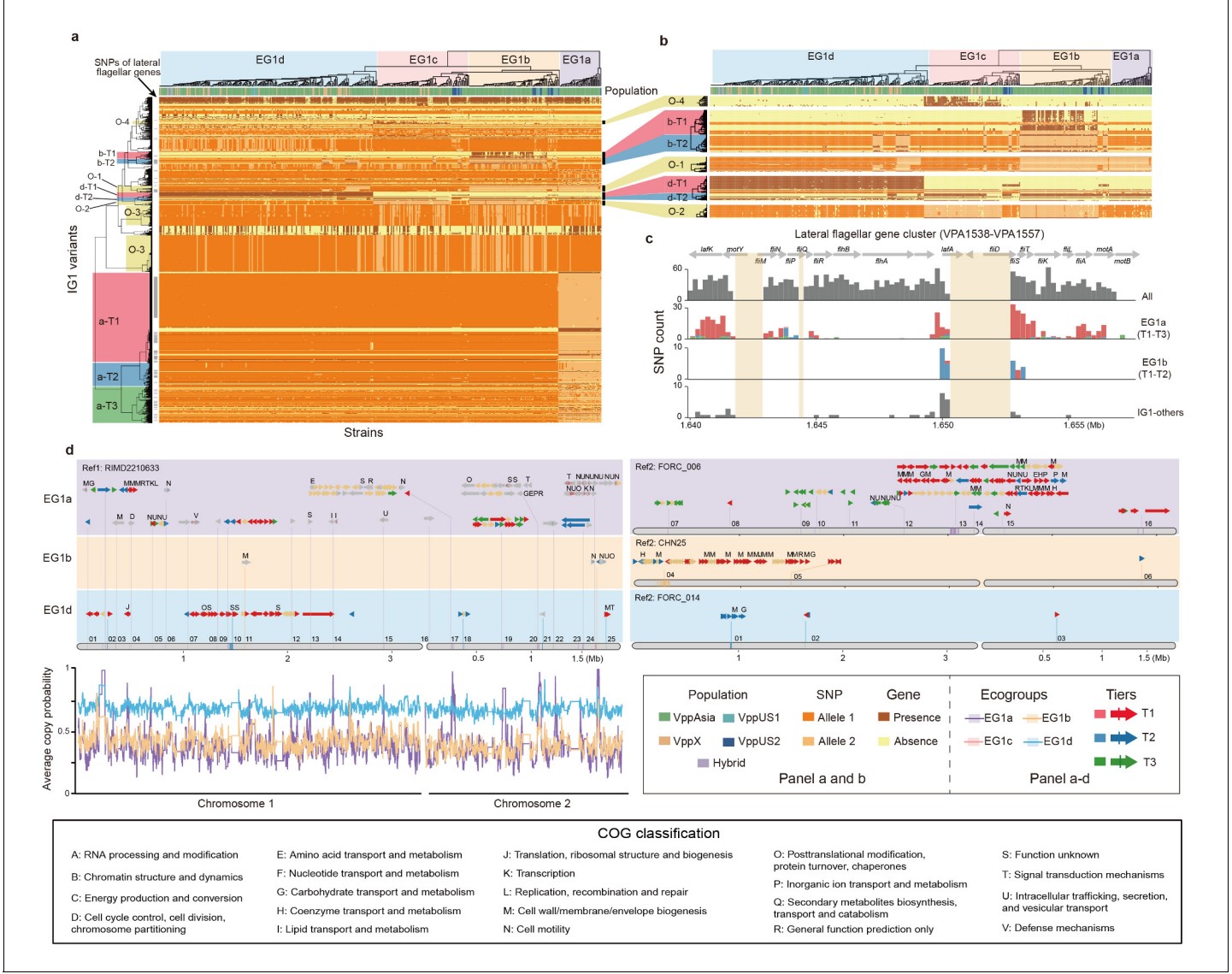

**Figure 3.** The largest interaction group (IG1) in *V. parahaemolyticus*. (a, b) Hierarchical clustering of 469 non-redundant strains (columns) based on coadaptated loci (rows) of IG1. Colors of the heatmap indicate the status of genetic variants as shown in the legend (bottom right), background colors of the upper and left clustering tree separately indicate different ecogroups (EG) and tiers, respectively, matching the colors in panel b-d. Colored bars below the upper clustering tree indicate the populations of strains. SNPs of lateral flagellar genes were marked by grey bars on the left of the heatmap. Panel b is a zoom-in version of specific tiers in panel a. (c) The distribution of coadaptation SNPs in the lateral flagellar gene cluster region (VPA1538-1557). The top indicates the gene organization of lateral flagellar gene cluster. Light orange rectangles indicate regions in the accessory genome. The histograms indicate the distribution of SNPs along the gene cluster, with colors of bars indicating coadaptation tiers. (d) Coadaptation blocks of IG1, shown in their genomic locations. Four different reference genomes were used, as indicated on the top of each panel, since no single genome contains all of the accessory genome variants. Grey horizontal arrows indicate core genes. Accessory genes are colored according to their tier or are tan if they do not belong to one. Vertical colored bars within grey arrows indicate core-SNPs in IG1, with colors indicating different coadaptation tiers. COG classification labels are shown above the genes. The numerical labels above reference genomes indicate the identity of the coadaptation genome blocks, corresponding to the information in *Supplementary file 1*. Colors of bottom left curves indicate the average copying probability (probability of genetic variants inherited from the same ecogroup) of different EGs calculated using chromosome painting.

The online version of this article includes the following figure supplement(s) for figure 3:

**Figure supplement 1.** Hierarchical clustering of 469 non-redundant strains (columns) based on different subsets of IG1 variants (rows).

**Figure supplement 2.** Neighbor-joining trees (top) and average copy probability value distributions (bottom) of geographical populations (a) and semi-clonal group (SCG) strains (b) of *V. parahaemolyticus*.

**Figure supplement 3.** NJ tree of lateral flagellar gene cluster region (VPA1538-1557) in *Vibrio* genus.

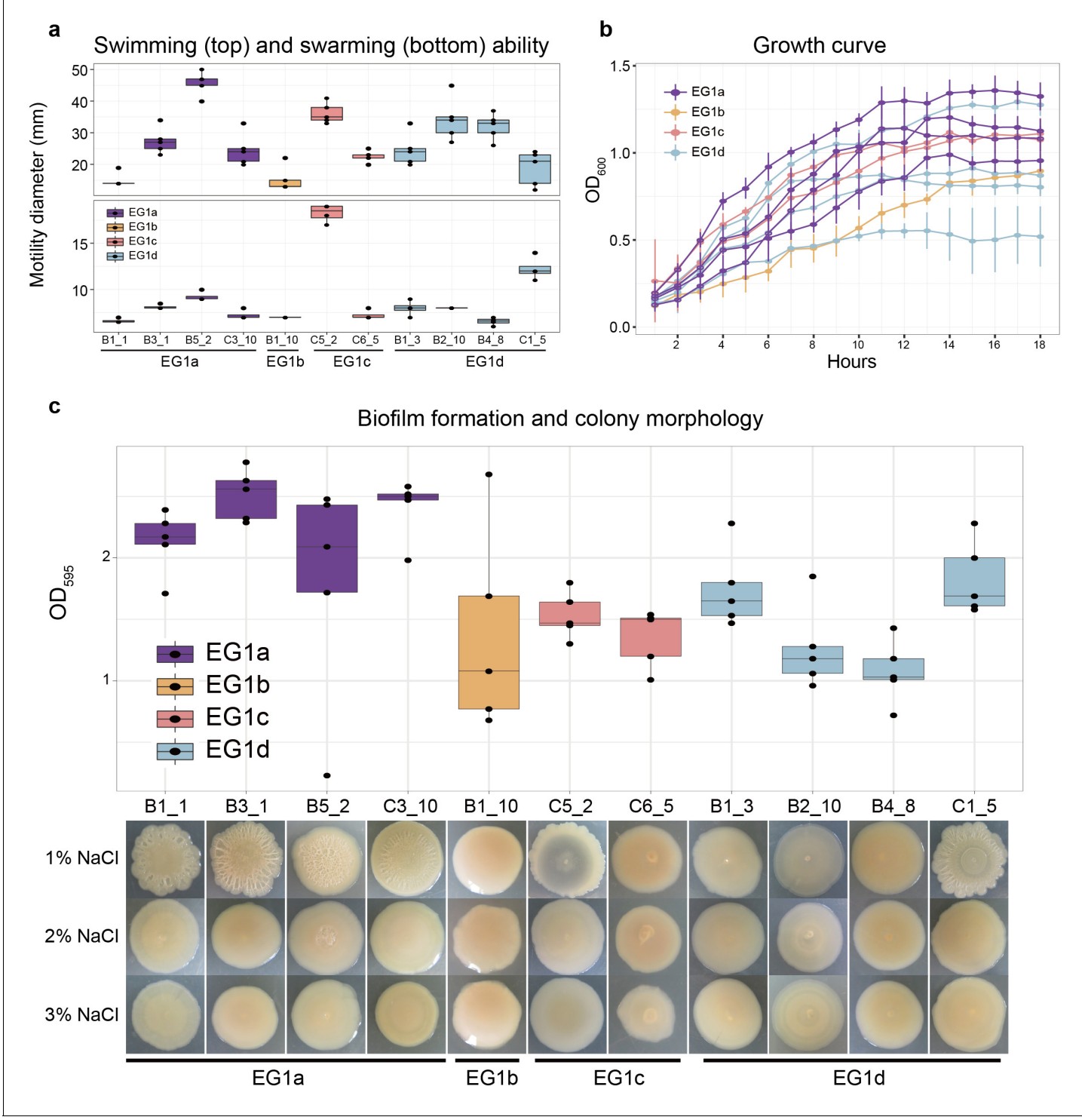

**Figure 4.** Phenotypes of strains from different EGs. (a) Swimming (top) and swarming (bottom) ability of strains in different ecogroups. Motility diameters in swimming and swarming plate were used to measure the motility ability. (b) Growth curve of strains in different ecogroups. The average optical density at 600 nm ($OD_{600}$) of five replicates were used to generate the curve, vertical lines indicate the standard deviation. (c) Biofilm formation (top) and colony morphology (bottom). $OD_{595}$ values were used to measure the biofilm formation ability. Colony morphologies of strains at different salinities were shown on the bottom. In panel a-c, five replicates were performed for each strain, and colors indicate different ecogroups.

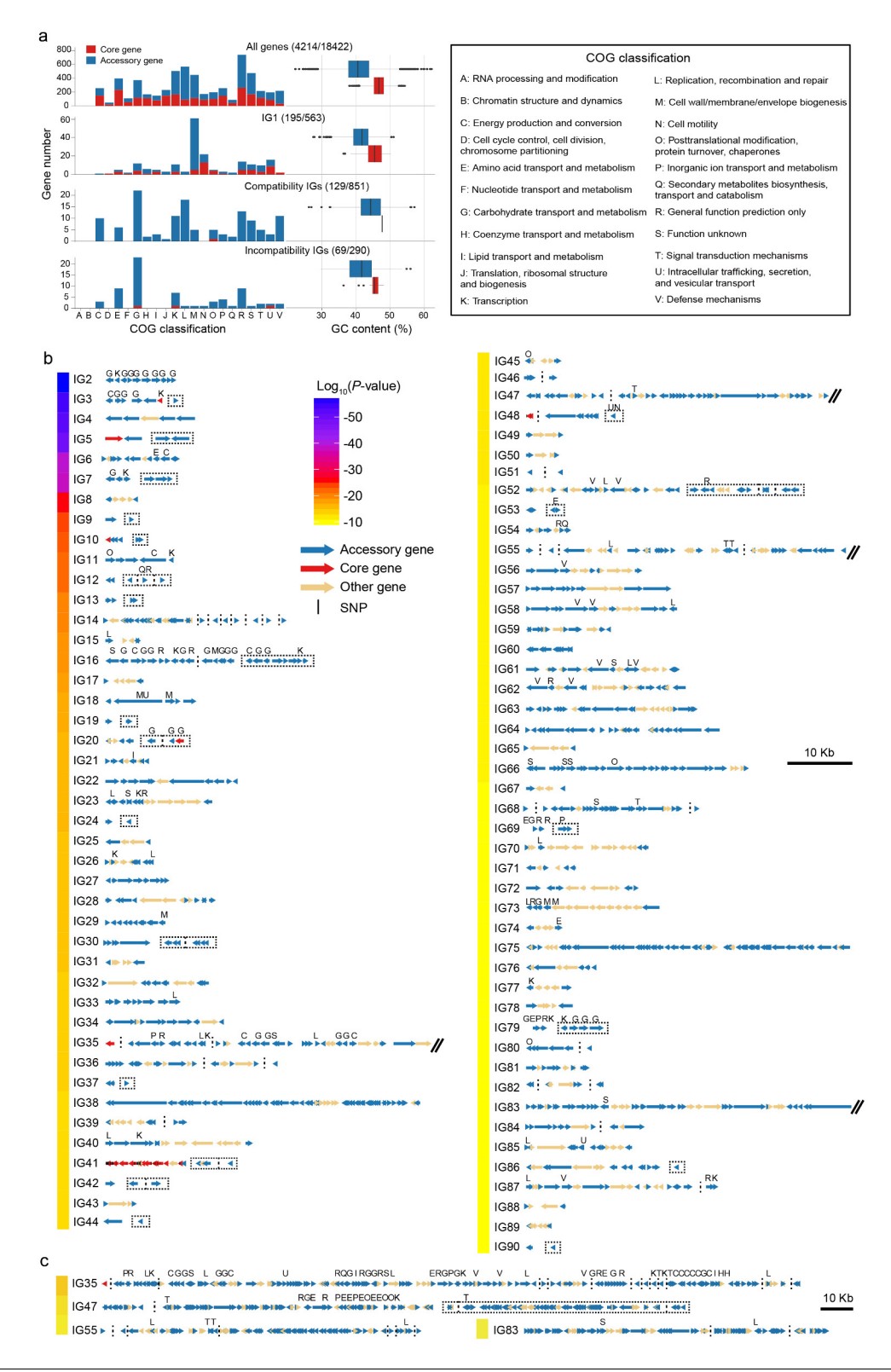

**Figure 5.** Characteristics of detected interaction groups. (a) COG classification and GC content of all analyzed genes (top panel) and of different types of coadapted genes (lower panels). Red for core genes and blue for accessory genes. The first number in brackets is the number of genes with COG annotation and the second is the total number of genes in the category. (b) Gene maps of different IGs. The colors of the bar on the left indicates average linkage strength of the loci in each IG. Arrows indicate genes. Acessory genes are shown in blue and genes with no coadaptation signal are

*Figure 5 continued on next page*

*Figure 5 continued*

shown in orange. Red genes indicated core genes containing coadapted SNPs which are indicated with black vertical lines. Vertical dotted lines were used to split compatible genes with physical distance larger than 3 kb, or genes located in different contigs, chromosomes and strains. Dotted rectangles indicate incompatible genes. IGs with genome block length larger than 60 kb are broken by double slash and shown in (c) after zooming out. COG classification labels are shown above the genes.

The online version of this article includes the following figure supplement(s) for figure 5:

**Figure supplement 1.** Clustering of 469 strains based on variations of IG2-90.

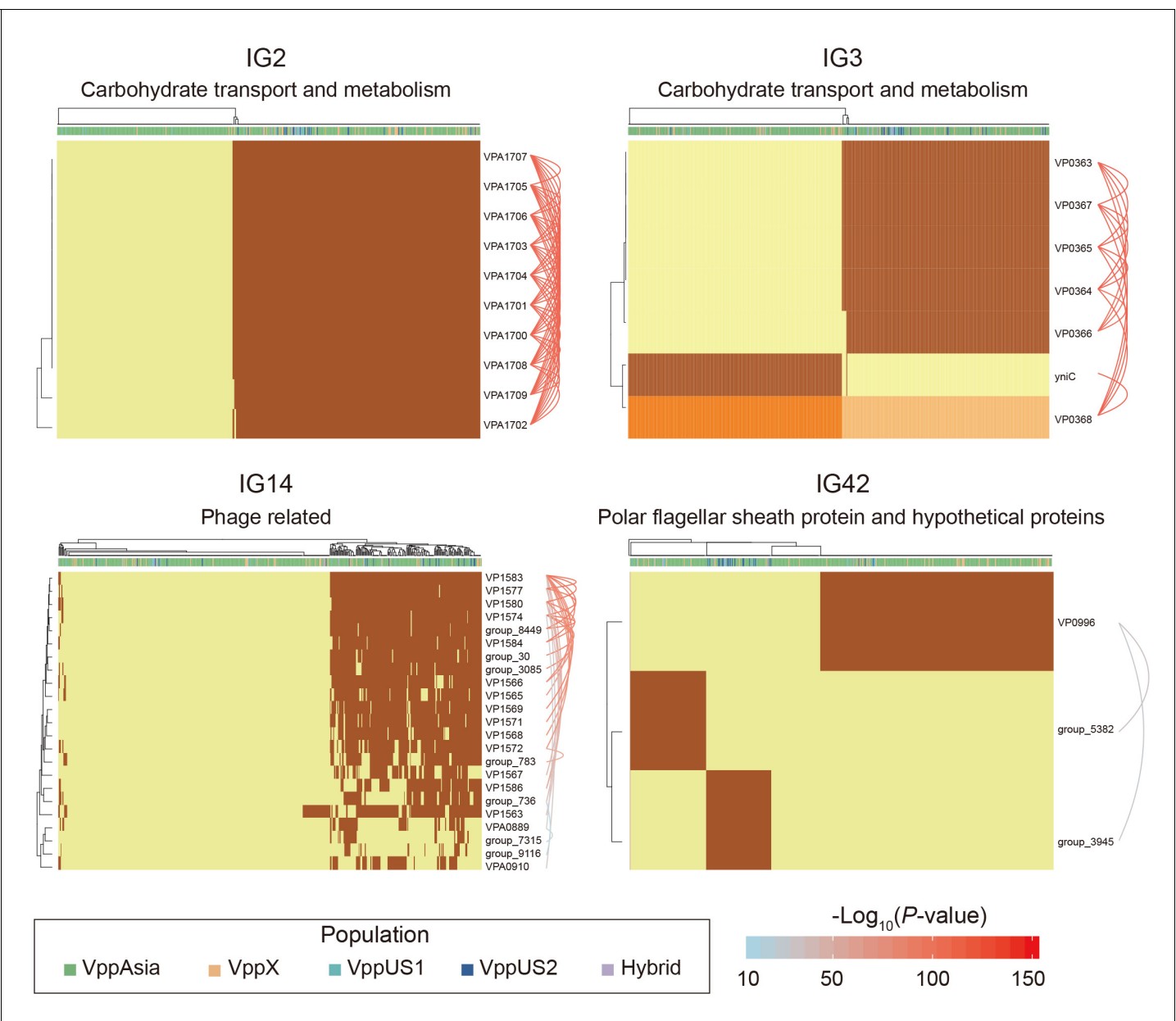

**Figure 6.** Four representative interaction groups. Hierarchical clustering of 469 non-redundant strains (columns) based on coadaptation loci (rows) of 4 representative IGs. Colors of the heatmap indicate the status of genetic variants, with light orange/orange for two alleles of a SNP, light yellow/brown for absence and presence of the accessory genes. Bar colors below the upper tree indicate the populations of strains according to the legend. A summary of the function of the involved genes is shown on the top of each heatmap. Arcs on the right indicate the causal links after ARACNE filtering, colors and the width of the arcs scale with the *P* values.

**Table 2.** Summary of interaction group 1 variants.

| Tier | Core SNPs | Accessory genes |
|---|---|---|
| a-T1 | 520 SNPs (359 Syn, 137 NonSyn) in 21 genes of 8 blocks, 14 genes encoding lateral flagellar | 66 genes in 10 blocks, 11 COG M genes, 5 T2SS genes |
| a-T2 | 121 SNPs (68 Syn, 43 NonSyn) in 18 genes of 10 blocks, 10 genes encoding lateral flagellar | 22 genes in 10 blocks, 2 COG M genes, 3 T2SS genes |
| a-T3 | 190 SNPs (137 Syn, 38 NonSyn) in 44 genes of 18 blocks, 17 genes encoding lateral flagellar | 46 genes in 13 blocks, 5 COG M genes, 3 COG NU genes, 4 T2SS genes |
| b-T1 | 12 SNPs (5 Syn, 2 NonSyn) lin 3 genes of 2 blocks, 2 genes encoding lateral flagellar | 33 genes in 2 blocks, 12 COG M genes |
| b-T2 | 25 SNPs (15 Syn, 10 NonSyn) in 3 genes of 1 block encoding lateral flagellar | 8 genes in 2 blocks, 1 COG M gene |
| d-T1 | 5 SNPs (3 syn, 2 nonsyn) in 1 gene encoding transmembrane | 31 genes in 5 blocks, 23 genes (1 block) encoding T6SS |
| d-T2 | 0 SNP | 14 genes in 4 blocks, 8 genes (1 block) encoding cellulose synthase |
| O-1 | 36 SNPs (28 Syn, 10 NonSyn) in 3 genes of 2 blocks, 2 genes encoding lateral flagellar, 1 *TonB* gene | 6 genes in 4 blocks, 1 COG H gene |
| O-2 | 27 SNPs (18 Syn, 1 NonSyn) in 3 genes in 1 block, 2 genes encoding *LuxR* family transcriptional regulator | 2 genes in 1 block, COG M and T |
| O-3 | 368 SNPs (269 Syn, 97 NonSyn) in 4 genes of 2 blocks, encoding multidrug resistance protein, lipase and long-chain fatty acid transport protein | 0 gene |
| O-4 | 0 SNP | 7 genes in 5 blocks, 4 genes encoding transferase |
| Others | 236 SNPs (182 Syn, 41 NonSyn) in 48 genes, 8 genes encoding lateral flagellar | 102 genes, 15 COG M genes |

## Functional differentiation of IG1 ecogroups

Within IG1, genes of COG classes M and N, cell wall biosynthesis and cell motility, are substantially overrepresented, relative to their overall frequency in the genome (COG M 32% vs 8%, COG N 13% vs 3%, $P<0.01$, Fisher exact test). 27% (509/1879) of the IG1 SNPs localize to the lateral flagella gene cluster (VPA1538-VPA1557). The variants are scattered widely in the heatmap in many different configurations, with Tier 1 and Tier 2 SNPs from lateral flagella gene cluster showing strong associations with EG1a (a-T1 and a-T2 variants) and EG1b (b-T1 and b-T2 variants) and other patterns (O-1 variants, *Figure 3a*, *Table 2*, *Supplementary file 1*). Consistent with this, hierarchical clustering based on the IG1 SNPs of lateral flagellar genes can also distinguish EG1a and EG1b strains (*Figure 3—figure supplement 1c*). The lateral flagellar locus is responsible for motility on surfaces and the different configurations, most involving multiple non-synonymous SNPs, suggests that there are several functionally distinct motility phenotypes encoded by different versions of the gene cluster. A phylogenetic analysis of variation at the lateral flagellar locus incorporating data from *Vibrio* genus shows that the locus has been inherited vertically within the species but with a substantially elevated rate of evolution in the EG1a version of the locus (*Figure 3—figure supplement 3*), which suggests that this version of the cluster has evolved a substantially new function.

EG1a strains are highly diverged compared to other EGs in the number of core SNPs (831 SNPs in 47 genes) and accessory genes (134 genes), including 114 fixed or nearly fixed non-synonymous differences (a-T1 and a-T2) at the lateral flagellar gene cluster. Additionally, EG1a strain has a particularly complex polymorphic genomic block (reference genome 2, genome block 13, Ref2-13, 82 kb, *Figure 3d*, *Supplementary file 1*) with substantial variation in gene content between strains, which in total constitutes for a large fraction of the gene content difference (60 genes). Among this genomic block, 14 genes were cell wall biosynthesis genes (COG M) and 5 genes are annotated as having Type II secretion related function.

EG1b strains are also associated with a large block, containing 34 accessory genes (Ref2-04, *Figure 3d*, *Supplementary file 1*), as well as a smaller number of lateral flagellar gene cluster SNPs

and only one further accessory gene and SNPs in a single gene, 12 of the accessory genes are cell wall biosynthesis genes (COG M). EG1c has no clear defining features, there are 7 genes (Tier O-4, *Supplementary file 1*) that are found at high frequency (67%) in EG1c while being rare elsewhere (5%), but none come close to representing a fixed difference. Four of these genes are annotated as being transferases.

Tier O-1 includes 16 SNPs in lateral flagellar genes (VPA1548 and VPA1550) and 20 SNPs in the gene *TonB* (VP0163), which was associated with the energy transduction (*Kuehl and Crosa, 2010*). These SNPs are strongly associated, despite being on different chromosomes, suggesting a specific functional interaction. Tier O-2 includes 18 SNPs in two core genes encoding *LuxR* family transcriptional regulators (VPA1446-VPA1447), which is associated with c-di-GMP signaling and biofilm formation (*Ferreira et al., 2012*). Within Tier O-2, ARACNE identified an accessory gene group_3560 as being a driver locus, with a particularly large number of causal interactions. This gene codes for a polysaccharide biosynthesis/export protein and retains interactions with 9 genome blocks after filtering (*Supplementary file 1*). Tier O-3 contains SNPs that are at high frequency in strains containing the group_3560 gene but are rare in the rest of the dataset. The SNPs are in four genes coding multidrug resistance protein (VP0038), hypothetical protein (VP0039), lipase (VPA0859) and long-chain fatty acid transport protein (VPA0860) respectively.

EG1d was identified in our previous analysis (*Cui et al., 2015*). In the current dataset, it is differentiated from other strains at 5 SNPs in a transmembrane gene (VPA1081) and 45 accessory genes. A 23 gene block encoding T6SS (type VI secretion system, VP1391-VP1420, Ref1-10, d-T1) is present or partly present in all EG1d strains, and absent or largely absent in the other three EGs. Most EG1a-c strains instead encode a block of genes that are annotated as being cellulose synthesis related (Ref2-01, d-T2). It is notable, however, that there are no SNPs associated specifically with EG1d in the lateral flagellar genome region.

## Phenotypic differences of EG1a strains

We performed a preliminary investigation of the phenotypic differences underlying EG1a by determining the motility, growth rate, and biofilm formation ability (*Figure 4*) of 11 strains (4 EG1a, 1 EG1b, 2 EG1c, 4 EG1d) on laboratory media. Because there are many unique variations in the lateral flagella gene cluster of EG1a isolates, we expected a different motility ability between EG1a and other isolates, but we failed to observe differences in swimming or swarming capability under the conditions tested (*Figure 4a*). However, EG1a strains revealed faster growth rate and significantly higher biofilm formation ability than EG1b-d strains (*Figure 4b,c*), and they revealed rough colony morphology, also an indication of increased biofilm formation, under low salinity (1% NaCl) culture condition (*Figure 4c*).

There are in total 60 EG1a strains in the global collection of 1,103 *V. parahaemolyticus* strains. All but one, a VppUS2 isolate, is from the VppAsia population, with the majority (n = 48, 80%) in this study coming from routine surveillance on food related environmental samples, including fish, shellfish, and water used for aquaculture. The strains revealed no clear geographical clustering pattern in China, as they can be isolated from all six provinces that under surveillance. Notably, only 4 EG1a strains were isolated from clinical samples, including wound and stool, representing a lower proportion than for EG1b-d isolates (453/1043), including if the two major pathogenic clonal lineages (CG1 and CG2) (*Yang et al., 2019a*) are removed (268/798), suggesting this ecogroup has low virulence potential in humans.

## Other interaction groups

The most common type of interaction group is 'genome island-like', which is a single accessory genome region of between 3 kb (IG50) and 57 kb (IG38), of which there are 52 (*Figure 5*). For example, IG2 consists of 10 genes in a single block (VPA1700-VPA1709). Nine of the genes code for various functions related to carbohydrate metabolism and transport while the 10[th] is a transcriptional regulator (*Figure 6* and *Supplementary file 2*). Four strains have 9 out of the 10 genes but otherwise the genes are either all present or all absent in every strain. Interestingly, there appears to be a difference in frequency between VppAsia isolates and others, with the island present in 52% of VppAsia strains and 90% of others.

Genome islands are often associated with transmission mechanisms such as phage and plasmids (*Dobrindt et al., 2004*). In our data, one example is IG14 which contains 4 genes annotated as being phage related and a further 19 hypothetical proteins (*Figure 6*). In the reference strain, 16 of the genes occur in a single block VP1563-VP1586, while 7 genes are present elsewhere in the genome. 56% of the 469 strains had none of the 23 genes, while 36% had more than 12 of them and 8% had between 1 and 5, which presumably represent remnants of an old phage infection. Only one gene (VP1563) in IG14 was found in appreciable frequency in strains that had none of the other genes. This gene might represent cargo of the phage infection that is able to persist for extended periods in the absence of infection due to a useful biological function of its own. Alternatively, it might additionally be transmitted by other mechanisms.

Another common interaction is incompatibility between different accessory genome elements, which is found in total 24 of the interaction groups (*Figure 5*). For example, in IG3 (*Figure 6*), all of the 469 strains either have the gene *yniC*, which is annotated as being a phosphorylated carbohydrates phosphatase, or at least 4 out of a set of 5 genes VP0363-VP0367 that includes a phosphotransferase and a dehydrogenase in the same genome location. Only one strain has both sets of genes. This interaction also involves a core genome SNP, in the adjacent gene VP0368, which is annotated as being a mannitol repressor protein. Another pattern is found in IG42 (*Figure 6*), where there are three genes (polar flagellar sheath protein and hypothetical proteins) that are mutually incompatible in our data, and 12% strains have none of the three genes.

Amongst interaction groups other than IG1, class G, encoding carbohydrate transport and metabolism are substantially overrepresented (22% vs 6%, p<0.01, Fisher exact test, *Figure 5a*), particularly amongst groups involving incompatibilities. There are also differences in GC content between accessory genes in IGs and others, with IGs having higher mean values, especially in compatibility IGs (*Figure 5a*).

## Comparison with other epistasis detection methods

We compared our results for core genome interactions with those obtained by SuperDCA, a method that uses Direct Coupling Analysis to identify causal interactions (*Morcos et al., 2011*; *Puranen et al., 2018*), using the default settings for the algorithm. To make the results as directly comparable as possible we used the same 198 isolates that were used for the Fisher exact test. The most important discrepancy is that although SNPs associated with a specific group of strains, EG1a, involves perfect associations, with $P$ values as low as $1.4 \times 10^{-30}$, the coupling strengths are lower than for the other groups we identified and none appear amongst the top 5,000 couplings (*Figure 7a*, *Figure 7—figure supplement 1*). This discrepancy is due the large number of SNPs involved with similar association patterns, which means that coupling values are distributed between them. Excluding EG1a SNPs, there is a strong correlation between SuperDCA coupling strengths and Fisher exact test $P$ values (*Figure 7a*). At a stringent cutoff of $10^{-2.2}$, SuperDCA identifies the same multi-locus interactions as Fisher exact test does at $P < 10^{-10}$, with a few SNPs excluded (*Figure 7b*). The significance thresholds for both the Fisher exact test and SuperDCA could be relaxed to identify a substantially larger number of true-positive hits, at the likely expense of some false ones, but we do not investigate these associations further here. We also compared our results with those obtained by SpydrPick, a model-free method based on mutual information (MI) (*Pensar et al., 2019*). The Fisher exact test $P$ value is almost perfectly correlated with the MI statistic used by SpydrPick for this data (*Figure 7a*).

## Discussion

Bacterial traits such as pathogenicity, host-specificity and antimicrobial resistance naturally attract human attention, but less obvious or even cryptic traits might be more important in determining the underlying structure of microbial populations. Individual strains assemble a set of variants from the gene pool of the population that allow them to overcome the challenges involved in colonizing specific niches. Genomics makes it possible to characterize both the overall genetic variation within the population and which sets of variants co-occur together, and which do not. Each combination of coadapted variants represents a distinct evolutionary solution to dealing with the multifaceted challenges that bacteria experience in finding viable niches to colonize. Therefore, studies of coadaptation provide a unique opportunity to see the world from the point of view of a bacterium.

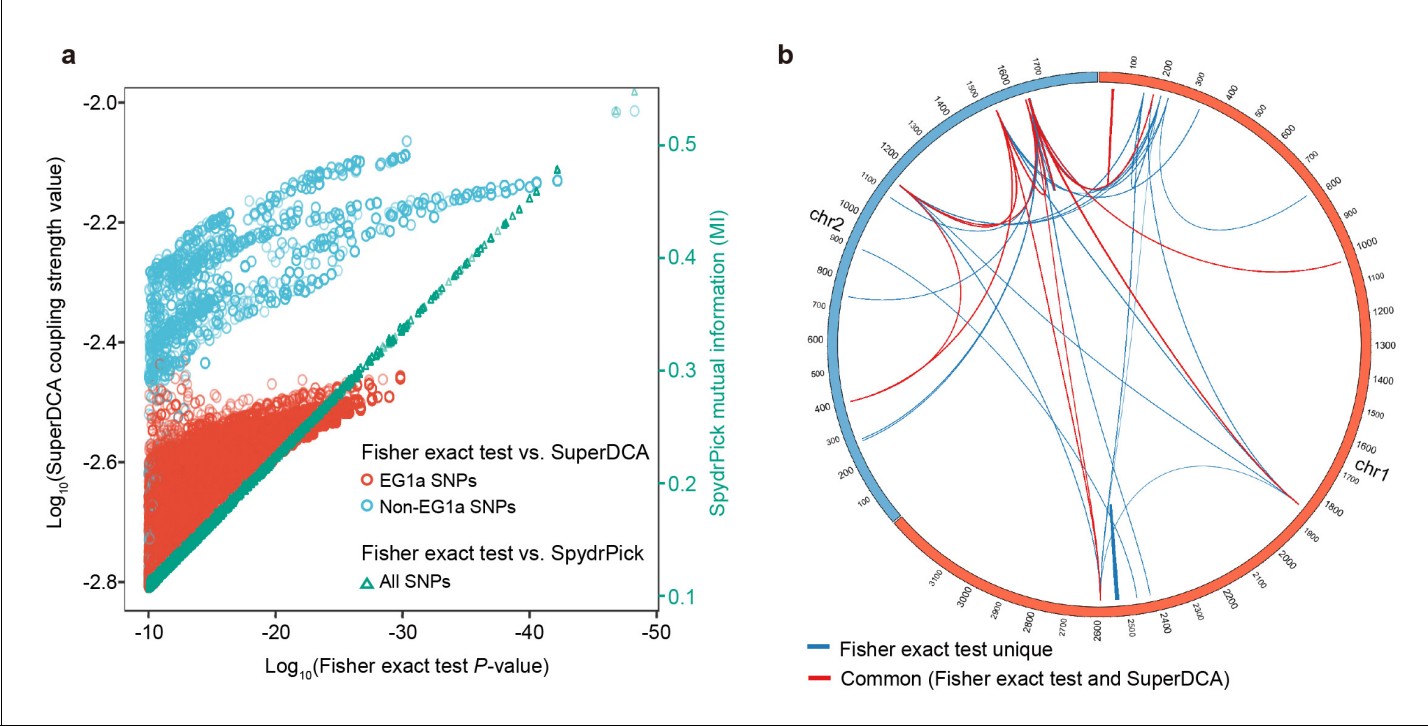

**Figure 7.** Comparison with other epistasis detection methods. (**a**) Correlation between Fisher exact test $P$ value and SuperDCA coupling strength (red and blue for ecogroup 1a (EG1a) and non-EG1a SNPs, respectively), and between Fisher exact test $P$ value and SpydrPick mutual information (green). (**b**) Overlap of strong linked SNP sites detected by Fisher exact test (p<$10^{-10}$, excluding EG1a SNPs) and SuperDCA (coupling strength >$10^{-2.2}$). Red for interacted SNP pairs detected by both methods, blue for SNP pairs detected only by Fisher exact test.

The online version of this article includes the following figure supplement(s) for figure 7:

**Figure supplement 1.** EG1a interactions that only detected by Fisher exact test.

We performed a genome wide scan for coadaptation in *V. parahaemolyticus*, performing pairwise tests for interactions amongst genetic variants and then clustered the significant pairwise interactions into 90 interaction groups. Because we focused on VppAsia population when identifying the coadapted genetic elements, we will not have detected genetic strategies any that are specific to non-VppAsia populations. We found that ecogroups found in VppAsia were always shared with at least one other population, but a more comprehensive coadaptation landscape of *V. parahaemolyticus* and its relationship to geography and other features will require more extensive sampling.

Our analysis demonstrated that genome wide epistasis scans can be used successfully to identify diverse interactions involving both core and accessory genomes but also highlighted unsolved methodological challenges. First, pairwise tests should, at least in principal, have reduced statistical power compared to methods that analyze all of the data at once, such as Direct Coupling Analysis (DCA) (*Morcos et al., 2011*). However, while there was a strong correlation between DCA and our results for core-genome interactions, DCA failed to identify the clearest, most extensive interaction in our dataset, namely the SNPs associated with EG1a. DCA was designed to identify coupling interactions that take place during protein folding and implicitly entails that pairwise interaction between a given pair of sites make it less likely that other sites will interact with either of them. For many types of interaction, a prior that makes the opposite assumption seems more appropriate, for example because master regulation loci are likely to interact with many different sites. Thus, in order to develop statistical tests that exploit the full power of genomic data, new types of statistical test that search for a more diverse range of interactions would need to be developed.

Second, distinguishing direct associations – either through gene function or ecology – from those that arise due to mutual correlation with other interacting genes, is a substantial, and largely unaddressed challenge. For the complex interaction groups in our data, the criteria used by ARACNE (*Margolin et al., 2006*) to remove interactions still left far too many interactions to be interpreted

usefully as being causal. We found that hierarchical clustering organized the interactions in a manner that allowed informal interpretation, but once again new statistical methodology is needed to facilitate detailed dissection of associations.

Notwithstanding the unresolved challenges, our results highlight the central role of lateral motility in structuring ecologically significant variation within the species. We also find evidence that interactions move through progressive stages, analogous to differing degrees of commitment within human relationships, namely casual, going steady, getting married and moving out together (*Figure 8*).

## Most interactions between core and accessory genomes are casual

A recent debate about whether the accessory genome evolves neutrally (*Vos and Eyre-Walker, 2017*; *Andreani et al., 2017*; *Shapiro, 2017*) highlighted how little we know about the functional importance of much of the DNA in bacterial chromosomes. Using the same statistical threshold to assess significance, our interaction screen identifies many more examples of coadaptation between different accessory genome elements than of interactions between the core and accessory genomes or within the core genome, implying that natural selection has a central role in determining accessory genome composition.

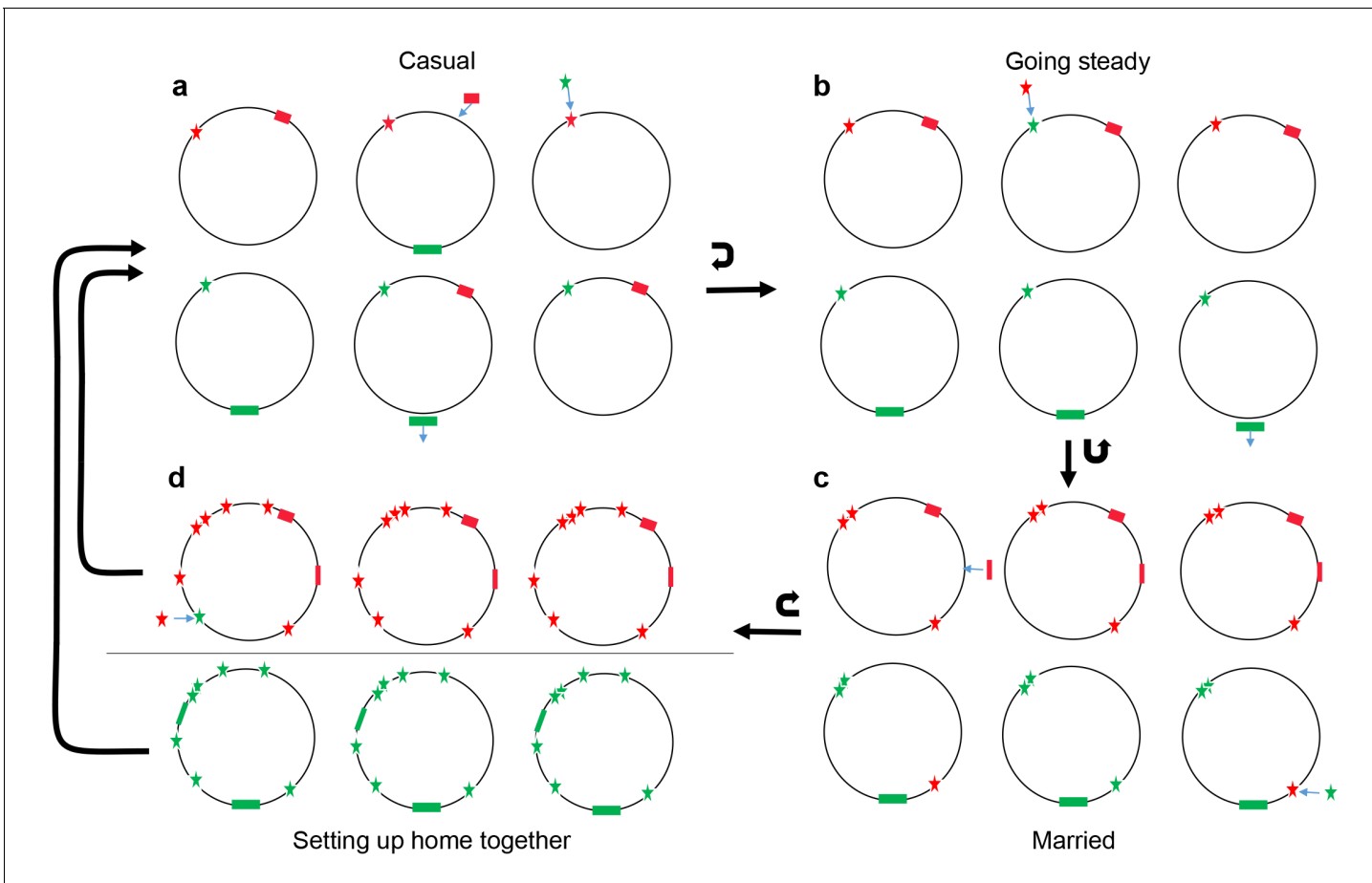

**Figure 8.** Conceptual model of four stages of coadaptation, analogous to human relationships. Circles indicate bacterial strains within a population. Red and green stars indicate the two alleles of a SNP. Red and green rectangles indicate different accessory genes or genome islands. Blue arrows indicate transitions between different SNP alleles, or gain/loss of genes and genome islands. Black arrows indicate the transitions between different stages, or cycles within a stage. (a) Casual: frequent gene flow generated multiple combinations of different variants within a population. Some combinations might have high fitness but alternate combinations arise frequently due to gene flux and adaptation at individual loci. (b) Going steady: coadapted interactions, such as the SNPs (red star) and accessory gene (red rectangles), become more difficult to dislodge despite ongoing genetic exchange due to their high fitness when present in combination. (c) Married: coadapted interactions become fixed in the population and led to further co-adaptation in multiple genome regions. (d) Setting up home together: as the progressive enlargement of coadapted regions in the genome, the entire genome becomes differentiated, which prompt the barriers (horizontal line) to genetic exchange between different ecogroups.

Unsurprisingly, given the extensive literature highlighting the importance of genomic islands to functional diversity of bacteria (*Dobrindt et al., 2004*), the most common form of adaptation detected by our screen is the coinheritance of accessory genome elements located in the same region of the genome. Based on a minimum size threshold of 3 kb, we find 52 (56%) such interactions, the largest of which is 57 kb (*Figure 5*).

Previous approaches to detecting genome islands emphasize traits associated with horizontal transmission, for example based on differences in GC content with the core genome, the presence of phage-related genes or other markers of frequent horizontal transfer (*Langille et al., 2010*). Our approach, based simply on co-occurrence identifies a wider range of coinherited units and suggests that many islands have functions related to carbohydrate metabolism.

Amongst interactions not involving physical linkage, the most common is incompatibility of different accessory genome elements, representing 27% (24/90) of our IGs. For example, two different versions of a phosphorous-related pathway, one involving one gene, the other involving 5 (IG3, *Figure 6*).

We propose that the rarity of interactions between core and accessory genome in our scan reflects the evolution of 'plug and play'-like architecture for frequently transferred genetic elements. Accessory genome elements are more likely to establish themselves in new host genomes if they are functional immediately on arrival in a diversity of genetic backgrounds. Furthermore, from the point of view of the host bacteria, acquisition of essential functions in new environments is more likely if diverse accessory genome elements in the gene pool are functional as soon as they are acquired. These selection pressures will tend to promote a genomic architecture that facilitates casual gain and loss of genetic elements from particular genomes (*Figure 8a*).

## Sometimes it makes sense to go steady

When an accessory genome element with an important protein coding function arrives in a new genome, it is likely that some optimization of gene regulation will be possible, coordinating the expression of the gene with others in the genome. This can lead to longer-lasting associations between genetic elements (*Figure 8b*). We found 8 different interaction groups involving core and accessory genome regions. One simple example included a regulatory gene VP0368 and an accessory genome element (IG3, *Figure 6*). In this example, it is feasible for the core genome SNP and the associated accessory element to be transferred together between strains in a single recombination event. Where coadaptation involves two or more separate genome regions, this makes assembling fit combinations more difficult and is likely to slow down the rate at which strains gain and lose the accessory genome elements involved. The difficulty of evolving the trait *de novo* is likely to slow down the rate at which it is gained and loss, which in turn makes further coadaptation at additional genes more likely.

IG1d, is an example of a complex coadaptation involving multiple core and accessory genome regions. A large majority of strains in our dataset (439/469, EG1d) either carry a cluster of genes encoding a T6SS, or a cluster encoding cellulose biosynthesis genes (EG1a-c), but few strains have genes from both clusters (*Figure 3a*). Cells uses the T6SS to inject toxins into nearby bacteria (*Salomon et al., 2013*) and cellulose production to coat themselves in a protective layer (*Tischler and Camilli, 2004*). Incompatibility might have a functional basis, for example because cellulose production prevents the T6SS functioning efficiently, or an ecological one, for example because cells that attack others do not need to defend themselves. The evolution of dissimilar strategies has led to differentiation in gene/SNP frequencies in a large number of regions. Strains that do not have the T6SS use at least three distinct sets of strategies, corresponding to ecogroups EG1a, EG1b and EG1c, encoded by alleles that are rare or absent in EG1d strains.

## Marriage changes everything

EG1a variants differ from the other interaction groups in our coadaptation screen in both the number of associated regions and the strength of the associations. The interaction group includes 454 SNPs in the lateral flagellar gene cluster (VPA1538-1557, 18 kb, *Figure 3c*), a further 917 core genome SNPs in 62 genes and 152 accessory genes in 35 clusters. Many of the variants represent fixed or nearly fixed differences between EG1a and other strains (a-T1 and a-T2). These include loci encoding flagellar genes, T2SS and other membrane transport elements. There are also 285 loci

(27%, a-T3) in weaker disequilibrium, typically because they are polymorphic in EG1a or the other EGs. Many of these variants are likely to represent more recently evolved coadaptations. Some of these genes are also associated with flagella or the T2SS related function but also encompass a broader range of functional categories, including cell division and amino acid transport and metabolism (*Table 2*, *Supplementary file 1*).

Our laboratory phenotype experiments (*Figure 4*) suggest that biofilm formation is likely to be a key trait underlying the different ecological strategies of EG1a and other EGs. However, the variation in phenotypic response at different salinity levels, and the absence of measurable difference in swarming behavior despite the large genetic difference within the lateral flagella genes, highlight some of the manifold difficulties of interpreting natural variation using phenotypes measured under laboratory conditions.

Despite the extensive differences that have accumulated between EG1a strains and the other EGs, there is no evidence of restricted gene flow in most of the genome (*Figure 3d*). Even within the flagellar gene cluster, strongly differentiated regions are separated by a weakly differentiated one (*Figure 3c*), implying that the coadaptation is being maintained by selection in the face of frequent recombination. Initial divergence in flagellar function is likely to have led to ecological differentiation, which led to bacteria for example having different nutritional inputs or requirements and a broadening of the functional categories undergoing divergent selection.

How can the difference between EG1a variants and the other interaction groups in both the number of associations and their strength be explained? *V. parahaemolyticus* is ubiquitous in shellfish in warm coastal waters, within which it occurs at densities of around 1,000 cells per gram, so a back of the envelope calculation suggests there are likely to be substantially more than $10^{15}$ bacteria in the VppAsia population. The species also has a high estimated effective population size (*Yang et al., 2018*; *Cui et al., 2015*) and has strong codon bias, which is often argued to be evidence that even tiny selective coefficients can drive adaptation (*Sharp et al., 2005*). Furthermore, recombination only breaks up linkage disequilibrium between loci slowly. Therefore, weaker and more variable patterns of association found for associations other than EG1a variants is unlikely to be a simple consequence of the ineffectiveness of selection and is instead likely to reflect complexity in the fitness landscape.

Strains gain flexibility by being able to switch between or modulate genomically encoded strategies by homologous recombination. The evolution of promiscuity, and plug and play-like architecture is self-reinforcing because the presence of strains using multiple strategies in the population also favors the presence of accessory genes and core gene haplotypes that have high or intermediate fitness on a wide range of different genetic backgrounds.

On the other hand, an absence of intermediate genotypes in the population can favor the evolution of fastidiousness, with particular accessory genes and haplotypes becoming essential components of some genetic backgrounds but deleterious on others. A likely scenario is that EG1a initially involved epistatic interactions between lateral flagellar variants and one or more additional loci elsewhere in the genome. Additional SNP differences accumulated making the two different versions of the lateral flagellar locus functionally more distinct. Greater functional distinctiveness of the two phenotypes in turn generated divergent selection pressures at additional loci, which led to further coadaptation, which made the phenotypes of the ecogroups still more distinct.

These arguments imply that the evolution of fastidiousness, like the evolution of promiscuity, can be self-reinforcing. At some point in the differentiation of EG1a strains from the remaining ones, the coadaptation of the core loci to each other is likely to have become more-or-less irreversible (*Figure 8c*). In this respect, the interaction between loci resembles the human institution of marriage, which has historically been thought of as an irreversible bond, although this notion loosened in the Christian world following the reign of King Henry VIII in England.

## Coadapted gene complexes as speciation triggers

Running the tape forward, it is easy to envisage the number of coadapted regions of the genome within EG1a undergoing progressive enlargement, until the entire genome becomes differentiated. As coadapted regions become more numerous, the proportion of recombination events between eco-groups that are maladaptive will increase, which might prompt the evolution of mechanistic barriers to genetic exchange between them (*Figure 8d*).

Mechanisms by which new bacterial species arise are frequently discussed in the literature (*Shapiro et al., 2012*; *Fraser et al., 2009*; *Falush et al., 2006*) but there is currently little data on how the process unfolds. EG1a isolates are of interest both as an example of an intermediate stage of divergence, prior to speciation, and because it suggests that substantial adaptive divergence between gene pools can precede any barriers to genetic exchange, other than natural selection at the loci involved. This – unique to our knowledge – example is exciting because the distinct signature of selection should make it possible to dissect the genetic basis of coadaptation in unprecedented detail. Broadly similar patterns of differentiation including 'genomic islands of speciation' have been observed for example between ecomorphs of cichlid fishes (*Malinsky et al., 2015*), but the evolution of ecomorphs has been facilitated by fish preferring to mate with similar individuals, which will have also inevitably lead to some level of differentiation at neutral loci throughout the genome.

## Conclusions

In *V. parahaemolyticus*, it has been possible to distinguish clearly between adaptive processes, reflecting fitness interactions between genes and neutral ones, reflecting clonal and population structure. This has allowed us to provide a description of the landscape of coadaptation, involving multiple simple interactions and a small number of complex ones. We have focused on interactions that generate strong linkage disequilibrium, but weaker and more complex polygenic ones also have the potential to provide biological insight.

Most bacteria have population structure that deviates more markedly from panmixia (*Yang et al., 2018*). In some species, this is likely due to smaller effective population sizes, lower recombination rates or mechanistic barriers to genetic exchange between strains. However, coadaptation can itself generate genome-wide linkage disequilibrium that might be difficult to distinguish from clonal or population structure. Because the linkage disequilibrium associated with IG1 is highly localized within the genome, it can, on careful inspection be clearly be attributed to selection, but in other bacteria patterns are likely to be less straightforward, making it challenging to understand to whether adaptive processes drive population structure, or vice versa. Natural selection is the jewel of evolution but distinguishing it from other processes requires in depth understanding of the relevant biology in addition to suitable data and statistical methods.

# Materials and methods

**Key resources table**

| Reagent type (species) or resource | Designation | Source or reference | Identifiers | Additional information |
|---|---|---|---|---|
| Software, algorithm | fineSTRUCTURE | http://paintmychromosomes.com https://doi.org/10.1371/journal.pgen.1002453 | SCR_018170 | |
| Software, algorithm | MUMmer | http://mummer.sourceforge.net/ https://doi.org/10.1186/gb-2004-5-2-r12 | SCR_018171 | |
| Software, algorithm | Prokka | https://github.com/tseemann/prokka https://doi.org/10.1093/bioinformatics/btu153 | SCR_014732 | |
| Software, algorithm | Roary | https://sanger-pathogens.github.io/Roary/ https://doi.org/10.1093/bioinformatics/btv421 | SCR_018172 | |
| Software, algorithm | TreeBest | http://treesoft.sourceforge.net/treebest.shtml | SCR_018173 | |
| Software, algorithm | iTOL | https://itol.embl.de/ https://doi.org/10.1093/nar/gkw290 | SCR_018174 | |
| Software, algorithm | SuperDCA | https://github.com/santeripuranen/SuperDCA https://doi.org/10.1099/mgen.0.000184 | SCR_018175 | |

*Continued on next page*

*Continued*

| Reagent type (species) or resource | Designation | Source or reference | Identifiers | Additional information |
|---|---|---|---|---|
| Software, algorithm | SpydrPick | https://github.com/santeripuranen/SpydrPick https://doi.org/10.1093/nar/gkz656 | SCR_018176 | |
| Software, algorithm | Circos | http://circos.ca/ https://doi.org/10.1101/gr.092759.109 | SCR_011798 | |

## Genomes used in this work

Totally 1,103 global *V. parahaemolyticus* genomes were used in this work, which also were analyzed in our other study (*Yang et al., 2019a*). To reduce clonal signals, we firstly made a 'non-redundant' dataset of 469 strains, in which no sequence differed by less than 2,000 SNPs in the core genome. They were attributed to 4 populations, VppAsia (383 strains), VppX (43), VppUS1 (18) and VppUS2 (21) based on fineSTRUCTURE result (*Lawson et al., 2012*). We then focused on VppAsia which has more strains, to generate a genome dataset in which strains represent a freely recombining population. We selected 386 genomes from 469 non-redundant genome dataset, including all the 383 VppAsia genomes and 3 outgroup genomes which were randomly selected from VppX, VppUS1 and VppUS2 population, respectively. These 386 genomes were used in Chromosome painting and fineSTRUCTURE analysis (*Lawson et al., 2012*) as previously described (*Cui et al., 2015*). Initial fineSTRUCTURE result revealed multiple clonal signals still exist, thus we selected one representative genome from each clone, combined them with the remaining genomes and repeated the process. After 14 iterations, we got a final dataset of 201 genomes with no trace of clonal signals, involving 198 VppAsia genomes that were used in further analysis (*Figure 1—figure supplement 1*).

The copying probability (probability of genetic variants inherited from the same population or ecogroup) value of each strain at each SNP was generated by Chromosome painting with '-b' option, and the average copying probability value of a given strain group (e.g. EG1a) at each SNP was used in *Figure 3d* and *Figure 3—figure supplement 2*.

## Variation detection, annotation and phylogeny

We re-called SNPs for 198 VppAsia genomes by aligning the assembly against reference genome (RIMD 2210633) using MUMmer (*Delcher et al., 2003*) as previous described (*Cui et al., 2015*; *Yang et al., 2019b*). Totally 565,466 bi-allelic SNPs were identified and 151,957 bi-allelic SNPs with minor allele frequency >2% were used in coadaptation detection. We re-annotated all the assemblies using Prokka (*Seemann, 2014*), and the annotated results were used in Roary (*Page et al., 2015*) to identify the pan-genome and gene presence/absence, totally 41,052 pan-genes were found and 14,486 accessory genes (present in >2% and<98% strains) were used in coadaptation detection. The pan-gene protein sequences of Roary were used to BLAST (BLASTP) against COG (*Galperin et al., 2015*) and KEGG (*Kanehisa and Goto, 2000*) database to obtain further annotation.

The Neighbour-joining trees were built by using the TreeBest software (http://treesoft.source-forge.net/treebest.shtml) based on sequences of concatenated SNPs, and were visualized by using online tool iTOL (*Letunic and Bork, 2016*).

## Detection of coadapted loci

Totally 151,957 bi-allelic SNPs and 14,486 accessory genes identified from 198 independent VppAsia genomes were used in coadaptation detection by three methods. Firstly, we used Fisher exact test to detect the linkage disequilibrium of each SNP-SNP, SNP-accessory gene, and accessory gene-gene pair. Presence or absence of an accessory gene were considered as being alternate alleles. Each variant locus (SNP or accessory gene) has two alleles, major and minor, of which major represents the allele shared by majority of isolates. For each pair of loci X and Y, the number of combinations between $X_{major}$-$Y_{major}$, $X_{major}$-$Y_{minor}$, $X_{minor}$-$Y_{major}$, $X_{minor}$-$Y_{minor}$ were separately counted and used in the contingency table to calculate the Fisher exact test *P* value. It took 3 days to finish all the coadaptation detection in a computer cluster using 21 cores and 2 Gb memory.

We also used SuperDCA (*Puranen et al., 2018*) and SpydrPick (*Pensar et al., 2019*) to detect the coadaptation between SNPs, using the same subset of 198 strains to make the analysis as comparable as possible. SuperDCA is based on direct coupling analysis (DCA) model (*Morcos et al., 2011*), a model to predict protein residue-residue contact from multiple sequence alignment, which has recently been extended to genome-wide epistasis analysis within bacterial populations. The demanding computation requirements hampered the direct application of DCA methods in big genome dataset. SuperDCA introduced optimized parallel inference algorithm, which is much faster than earlier DCA methods and is scalable for up to $10^5$ variants. However, it still took 25 days to finish the detection by using 32 cores and 86 Gb memory.

SpydrPick is based on mutual information (MI), an information theoretic measure of mutual dependence between two variants. Pairwise analysis of variants using SpydrPick is scalable to core genome SNP and pan-genome-wide analysis. SpydrPick took one hour to finish the calculation using 32 cores and 1 Gb memory.

We removed coadaptation pairs with distance less than 3 kb to minimize the influence of physical linkage. All identified SNPs in this study were located in the core genome, therefore the physical distance between SNP pairs can be calculated according to their position in the reference genome. To define the distance between accessory genes, and between SNP and accessory gene, we mapped the sequence of accessory genes against available 19 complete maps of the *V. parahaemolyticus* genomes to acquire their corresponding position, and then the gene that failed to be found in complete reference genomes were then mapped to the draft genomes. If the accessory genes pair or SNP- accessory gene pair was found located in a same chromosome or same contig of a draft genome, then the distance between paired variants could be counted according to their position in the chromosome or contig. The distance between paired variants that located in different chromosomes or contigs was counted as larger than 3 kb and such pairs were kept in further analysis. Circos (*Krzywinski et al., 2009*) was used to visualize the networks of coadaptation SNPs in *Figure 7b* and *Figure 7—figure supplement 1*.

## Lateral flagellar gene cluster region in *Vibrio* genus

To identify the homologous sequences of *V. parahaemolyticus* lateral flagellar gene cluster (VPA1538-1557) in the *Vibrio* genus, we downloaded all available *Vibrio* genome assemblies in NCBI, then aligned the nucleotide sequence of lateral flagellar gene cluster of *V. parahaemolyticus* (NC_004605 1639906–1657888) against *Vibrio* genome dataset (excluding *V. parahaemolyticus*) by using BLASTN. Totally 46 *Vibrio* genomes revealed above than 60% coverage on lateral flagella region in *V. parahaemolyticus* genome and was used in phylogeny rebuilding. We also included three randomly selected strains from EG1a and EG1b-d respectively for comparison. In total 3,000 SNPs were identified in this region and were used for NJ tree construction.

## Determination of phenotypes

### Bacteria strains

In the phenotype experiments, totally 11 strains were randomly selected respectively from four EGs that defined by IG1 variants, including 4 EG1a strains (B1_1, B3_1, B5_3, C3_10), 1 EG1b strain (B1_10), 2 EG1c strains (C5_2, C6_5) and 4 EG1d strains (B1_3, B2_10, B4_8, C1_5). The strains stored at −80℃ were inoculated in the thiosulfate citrate bile salts sucrose agar (TCBS) plates by streak plate method. Five clones for each strains were inoculated again in another TCBS plate and then cultured overnight at 30℃ in 3% NaCl-LB broth overnight and used for the following assays.

### Motility assays

Five clones for each strain were cultured overnight at 30℃ and then inoculated in the swimming plate (LB media containing 0.3% agar) and swarming plate (LB agar with 3% NaCl). The swimming ability was recorded by measuring the diameter of colony after 24 hr at 30℃. And the swarming ability was recorded after 72 hr at 24℃.

### Growth curve

*V. parahaemolyticus* strains in 96-well plate were cultured overnight at 30℃ in 3% NaCl-LB broth. The optical density of each culture was adjust to an $OD_{600}$ of 0.6.Then 1 ml of each culture was

inoculated 100 ml of 3% NaCl-LB broth in a 96-well plate and cultured at 30℃. The growth of each culture were measured every 1 hr at the optical density of 600 nm using Multiskan Spectrum.

## Biofilm formation

*V. parahaemolyticus* strains were cultured overnight at 30℃ in 3% NaCl-LB broth. 2 μl of each overnight culture was inoculated to 100 μl of 3% NaCl-LB broth in a 96-well plate and cultured at 30℃ for 24 hr statically. The supernatant was discarded and each well was washed once with sterile phosphate-buffered saline (PBS). 0.1% Crystal violet (wt/vol) was added to each well and incubated at room temperature for 30 min. The crystal violet was decanted, and each well was washed once with sterile PBS. Crystal violet that stained biofilm was solubilized with dimethylsulfoxide (DMSO), and then measured at the optical density of 595 nm using Multiskan Spectrum (Thermo Scientific).

## Acknowledgements

This work is supported by the National Key Research and Development Program of China (No. 2017YFC1601503 and 2018YFC1603902), National Key Program for Infectious Diseases of China (No. 2018ZX10101003 and 2018ZX10714–002), Sanming Project of Medicine in Shenzhen (No. SZSM201811071), the National Natural Science Foundation of China (No. 31770001) and the Key Research Program of the Chinese Academy of Sciences (No. ZDRW-ZS-2017–1). DF is funded by a Medical Research Council Fellowship as part of the MRC CLIMB consortium for microbial bioinformatics (No. MR/M501608/1), Shanghai Municipal Science and Technology Major Project (No. 2019SHZDZX02).

## Additional information

### Funding

| Funder | Grant reference number | Author |
|---|---|---|
| National Key Research and Development Program of China | 2017YFC1601503 | Yujun Cui |
| National Key Research and Development Program of China | 2018YFC1603902 | Yujun Cui |
| National Key Program for Infectious Diseases of China | 2018ZX10101003 | Ruifu Yang |
| National Key Program for Infectious Diseases of China | 2018ZX10714-002 | Ruifu Yang |
| Sanming Project of Medicine in Shenzhen | SZSM201811071 | Yujun Cui Chao Yang |
| National Natural Science Foundation of China | 31770001 | Yujun Cui |
| Key Research Program of the Chinese Academy of Sciences | ZDRW-ZS-2017-1 | Hui Wang |
| Medical Research Council | MR/M501608/1 | Daniel Falush |
| Shanghai Municipal Science and Technology Major Project | 2019SHZDZX02 | Daniel Falush |

The funders had no role in study design, data collection and interpretation, or the decision to submit the work for publication.

### Author contributions

Yujun Cui, Conceptualization, Data curation, Formal analysis, Supervision, Funding acquisition, Validation, Investigation, Methodology, Writing - original draft, Project administration, Writing - review and editing; Chao Yang, Data curation, Software, Formal analysis, Validation, Investigation, Visualization, Methodology, Writing - original draft, Writing - review and editing; Hongling Qiu, Hui Wang,

Formal analysis, Validation, Methodology; Ruifu Yang, Conceptualization, Formal analysis, Supervision, Investigation; Daniel Falush, Conceptualization, Resources, Data curation, Formal analysis, Supervision, Funding acquisition, Validation, Investigation, Methodology, Writing - original draft, Project administration, Writing - review and editing

## Author ORCIDs
Chao Yang (iD) https://orcid.org/0000-0003-0626-0586
Daniel Falush (iD) https://orcid.org/0000-0002-2956-0795

## Decision letter and Author response
Decision letter https://doi.org/10.7554/eLife.54136.sa1
Author response https://doi.org/10.7554/eLife.54136.sa2

# Additional files

## Supplementary files
- Supplementary file 1. Detailed information of coadaptation variants in IG1.
- Supplementary file 2. Detailed information of coadaptation variants in IG2-90.
- Transparent reporting form

## Data availability
All data are publicly available.

The following previously published dataset was used:

| Author(s) | Year | Dataset title | Dataset URL | Database and Identifier |
|---|---|---|---|---|
| Yang C, Pei X, Wu Y, Yan L, Yan Y, Song Y, Coyle N, Martinez-Urtaza J, Quince C, Hu Q, Jiang M, Feil E, Yang D, Zhou D, Yang R, Falush D, Cui Y | 2019 | Vibrio parahaemolyticus Genome sequencing and assembly | https://www.ncbi.nlm.nih.gov/bioproject/393608 | NCBI BioProject, PRJNA393608 |

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
