## [Decision Letter]

**Acceptance summary:**

The work contributes significantly to a central problem, namely, seemingly disparate evolution of core and pan-genomes, but as the study shows, there are significant genetic associations. Additionally, the work provides insight into how sympatric speciation might occur in the face of rampant gene flow.

**Decision letter after peer review:**

Thank you for submitting your article "The landscape of coadaptation in Vibrio parahaemolyticus" for consideration by *eLife*. Your article has been reviewed by Dominique Soldati-Favre as the Senior Editor, a Reviewing Editor, and two reviewers. The reviewers have opted to remain anonymous.

The reviewers have discussed the reviews with one another and the Reviewing Editor has drafted this decision to help you prepare a revised submission.

This is an interesting and clever study assessing the extent and implication of pan-genome genetic associations at a species-wide scale. The study takes advantage of a fairly unique phylogenetic structure of Vibrio parahaemolyticus to eliminate the extremely challenging population structure issue, and by doing so enables what is really a conceptually straight-forward, genome-wide analysis of associations among genetic variation (both gene presence/absence and SNPs). The analysis revealed 90 interaction groups, including one that encompassed 1540 SNPs in 82 core and 338 accessory genes, many of which are associated with the lateral flagella and cell wall. But even more interesting is the relative paucity of associations between the core and accessory genomes. Finally, the authors propose a number of valuable hypotheses concerning how these associations come about and the consequences of the interactions for adaptation and divergence. Overall, the work is novel, illuminating and advances the field, particularly the microbial population genetics community because of the unique perspective on how sympatric speciation may initiate despite the presence of gene flow.

Essential revisions:

None of the reviewers had particularly major issues, but both, along with the BRE, agree that more needs to be done to help the reader traverse and remain connected to the Results. The figures and especially the legends need to be improved.

Additionally, we would like to see the fineSTRUCTURE analysis augmented by a simple PCoA (or MDS) done separately for both the core and accessory variants.

I have included all the comments from the referees and trust the authors will pay attention to these valuable and additional suggestions/requests when revising the manuscript.

*Reviewer 1:*

The figures are complex and only minimally explained. This is particularly true when trying to juggle IGs, EGs, O/a/b/c/d, tiers, locus IDs, strain designations. I'm sure these are all clear to the authors, but readers are seeing this for this first time and it's a bit of an alphabet soup. I will make some specific points below, but in general the figures need to be described more thoroughly and clearly.

There are quite a few statements about associations or enrichments that are not supported by numbers of stats. Please don't state there is an association without the supporting numbers and statistical analysis. I will point out a few of these below.

Could the focus on VppAsia have biased the results in any way? Do you think similar conclusions would be drawn if other subspecies are species are used?

Subsection “Detection and characterization of interaction groups”. A more explicit definition of the network would be useful. Are these single-linkage networks where only one significant connection is required for the inclusion of a variant into a network?

– What does "substantially enriched" mean. Some numbers and a simple statistical analysis would be helpful.

– You don't introduce EGs until quite a bit later in the manuscript, so at least spell out ecogroup to give the reader some clue as to what is going on.

Subsection “Detailed characterization of the largest interaction group”. "strong association" Please clarify.

– It might be useful to show the cluster analysis on just the lateral flagellar genes.

– What is Ref2-13? Same for similar notations elsewhere.

Subsection “Marriage changes everything”. While the references to Henry VIII is funny, I'm not sure that all members of the international science community will understand.

Subsection “Detection of coadapted loci”. A little more thorough description of SuperDCA and SpydrPick would be appreciated.

Figure 1. I don't think panel B is a Q-Q plot. It is really a frequency distribution of P-values. As mentioned for for subsection “Detection and characterization of interaction groups”, you haven't introduced EG1 yet, so minimally write out ecogroup.

Figure 4. Although you state it in the legend, it would help readers if you labeled the x- and y-axes in panel A with 'Strains' and 'Loci'. The whole nomenclature is pretty confusing and is indecipherable to a reader who is skimming the figures. You should repeat some of the description you make in the text (around subsection “Detailed characterization of the largest interaction group”). Why is panel B broken out? Explain this in the legend. I still don't fully understand panel D. You state that 4 reference genomes are shown. Presumably these correspond to the text such as RIMD2210633, but this is not stated anywhere. Why do you have Ref1 and then 3 different Ref2s next to these designations? How come the tan and blue blocks on the left side of panel 1 don't have a designation? Are they all from Ref1: RIMD221063? I don't understand how that could be. And what exactly is the copying probability? This figure legend needs a lot of work.

Figure 5. The legend is very non-descriptive and needs work.

Bottom line regarding these figure legends. I was so annoyed trying to work through the figures that I almost suggested rejecting the manuscript. Fortunately, I decided to put it aside for the day and come back with a cooler head. But don't make this so hard on readers.

*Reviewer 2:*

I would appreciate a more thorough description of the genome diversity used in different places in the study. Primarily, the interaction groups are identified using a very carefully curated set of strains without population structure, but analyses on the distribution of those SNPs and elements is made in a larger set that is not similarly characterized in the manuscript. I would like to see a similar figure for Figure 1A for the set of 469 strains used throughout the analysis.

Additionally, the star phylogeny from Figure 1A convinces me that there is no population structure present in the dataset used for identifying the IGs, but I have a poor intuition for how closely related the strains are because I am unsure how their SNP filtering could affect the tree produced. I think a maximum likelihood ribosomal gene tree with the scale in substitutions per site would better orient readers to the scale of diversity included in these analyses. This would be especially interesting to know how the strain groups from IG1 heirarchical clustering maps onto this tree.

Related to this point of understanding the genome diversity of these strains as how they relate to the distribution of IGs, I would be curious to see if the strain groupings from the hierarchical clustering of IG1 could be reproduced by hierarchical clustering of strains based on flexible genome content.

Regarding the analogy presented in Figure 6, it was not explained when it is introduced in the Discussion section and the figure legend for this figure is not detailed in its explanation of how to read the schematic so I am not confident in my understanding of the meaning of the various details included in it. I think the manuscript would benefit from a more explicit description of their model for how such complex interaction groups could come to exist in natural microbial populations.

Concerning making the results easier to follow, I recommend changing figure legends and sub-headings within the Results section to be more descriptive and convey the take-away message the reader should have from reading that figure or section. Similarly, summary and topic sentences throughout the results would make appreciating the implications of results easier.

Subsection “Detailed characterization of the largest interaction group” – "more-or-less strongly associated" please clarify your wording here.

Figure 4 legend – restate color scheme of heatmap in both Figure 3 and Figure 4 legends instead of referring to previous figure legend.

---

## [Author Response]

Essential revisions:None of the reviewers had particularly major issues, but both, along with the BRE, agree that more needs to be done to help the reader traverse and remain connected to the Results. The figures and especially the legends need to be improved.Additionally, we would like to see the fine STRUCTURE analysis augmented by a simple PCoA (or MDS) done separately for both the core and accessory variants.I have included all the comments from the referees and trust the authors will pay attention to these valuable and additional suggestions/requests when revising the manuscript.

We thank the editors and the reviewers for their constructive feedback which we think has considerably improved the manuscript. We have revised the manuscript, especially the figure legends to make it clear.

The suggestion of adding a PCoA analysis was an extremely valuable one, particularly from an expository point of view. We have added a largely new Figure 1 which contains phylogenetic trees and PCoA results for both the non-redundant dataset and the discovery dataset. The different PCoA analyses illustrate the effect of geographic population structure and ecogroup structure and therefore provides a very useful link connecting together all of the different analyses. IG1 ecogroups are clearly visible in the PCoA plots so we have restructured the results to discuss IG1 before the other interaction groups. We believe this restructuring has greatly improved the digestibility of the manuscript. We have clarified these in the main text as follows:

Subsection “Detection and characterization of interaction groups”: These four populations are clearly visible in Principle Coordination Analysis (PCoA) of the SNP variants in this dataset (Figure 1B). […] We then used fineSTRUCTURE (12) to look for signals of clonal structure within the remaining isolates, iteratively removing isolates until all of the isolates were assigned to a single population (Methods), leading to a discovery dataset of 198 strains (Figure 1C, Figure 1—figure supplement 1).

Subsection “Complex structure of interaction group 1”: Although fineSTRUCTURE did not detect any population structure within the discovery dataset, PCoA analysis of SNPs identifies axes of variation that reflect the ecogroup structure within EG1 (Figure 1d). […] Geographically based population structure is correspondingly less important, with weak differentiation of two of the populations (VppAsia and VppX) from the other two evident within PCo2 (Figure 1B, D), which is consistent with our previous observation that few population-specific genes were identified (Yang et al., 2019).

Reviewer 1:The figures are complex and only minimally explained. This is particularly true when trying to juggle IGs, EGs, O/a/b/c/d, tiers, locus IDs, strain designations. I'm sure these are all clear to the authors, but readers are seeing this for this first time and it's a bit of an alphabet soup. I will make some specific points below, but in general the figures need to be described more thoroughly and clearly.

We are sorry for these failings. The figure legend and the main text now carefully introduces the definition of IGs, EGs, tiers etc., which are explained the first time they are introduced.

There are quite a few statements about associations or enrichments that are not supported by numbers of stats. Please don't state there is an association without the supporting numbers and statistical analysis. I will point out a few of these below.

We have provided supporting numbers and statistical analysis for all associations now.

Could the focus on VppAsia have biased the results in any way? Do you think similar conclusions would be drawn if other subspecies are species are used?

We agree that the picture is likely to be incomplete (rather than necessarily biased, since it is possible just to consider the VppAsia population in isolation) and now highlight this limitation in the Discussion section.

Discussion section”: Because we focused on VppAsia population when identifying the coadapted genetic elements, we will not have detected genetic strategies any that are specific to non-VppAsia populations. We found that ecogroups found in VppAsia were always shared with at least one other population, but a more comprehensive coadaptation landscape of *V. parahaemolyticus* and its relationship to geography and other features will require more extensive sampling.

Subsection “Detection and characterization of interaction groups”. A more explicit definition of the network would be useful. Are these single-linkage networks where only one significant connection is required for the inclusion of a variant into a network?

Only one significant connection was required, which we have clarified. We removed the term “network” since it represents an unnecessary duplication of terms with “Interaction Group”. and revised the text as follows:

Subsection “Complex structure of interaction group 1”: This left us with 452,849 interactions with *P*<10^-10^, which grouped into 90 interaction groups (IG, a set of SNP/accessory gene connected by at least one significant interaction), all of which involved at least one accessory-genome element…

– What does "substantially enriched" mean. Some numbers and a simple statistical analysis would be helpful.

Thank you for pointing out this. We have provided supporting numbers and statistical analysis in current version.

Subsection “Complex structure of interaction group 1”: Interacting SNPs were enriched for non-synonymous variants (26% vs 16%, *P* < 0.01, Fisher exact test), which is consistent with natural selection…

– You don't introduce EGs until quite a bit later in the manuscript, so at least spell out ecogroup to give the reader some clue as to what is going on.

As part of the reorganization of the manuscript described above, we now describe EGs before comparing with other methods.

Subsection “Complex structure of interaction group 1”: However, hierarchical clustering revealed that strains fall into four distinct “ecogroups” (EGs) based on IG1 variants.

Discussion section: The most important discrepancy is that although SNPs associated with a specific group of strains, EG1a, involves perfect associations…

Subsection “Detailed characterization of the largest interaction group”. "strong association" Please clarify.

We have explained that “strong association” means Tier 1 and Tier 2 variants (a-T1, a-T2, b-T1, b-T2 and O-1 variants), and revised the text to clarify this as follows:

Subsection “Functional differentiation of IG1 ecogroups”: The variants are scattered widely in the heatmap in many different configurations, with Tier 1 and Tier 2 SNPs from lateral flagella gene cluster showing strong associations with EG1a (a-T1 and a-T2 variants) and EG1b (b-T1 and b-T2 variants) and other patterns (O-1 variants, Figure 3A, Table 2, Supplementary file 1).

– It might be useful to show the cluster analysis on just the lateral flagellar genes.

Thank you for this suggestion. We performed hierarchical clustering analysis based on the IG1 SNPs of lateral flagellar genes (Figure 3—figure supplement 1). The clustering results of ecogroup (EG) and tier were similar with the assignment based on all the IG1 variant. Because the presence of EG1a and EG1b specific SNPs (a-T1~T3, b-T1~2), EG1a and EG1b strains generally clustered together, but EG1c and EG1d strains were mixed. We have added the figure as Figure 3—figure supplement 1C and clarified this in the main text as follows:

Subsection “Functional differentiation of IG1 ecogroups”: Consistent with this, hierarchical clustering based on the IG1 SNPs of lateral flagellar genes can also distinguish EG1a and EG1b strains (Figure 3—figure supplement 1C).

– What is Ref2-13? Same for similar notations elsewhere.

Ref2-13 indicated the genome block 13 in the reference genome 2. We have clarified this in the main text and figure legend now.

Subsection “Functional differentiation of IG1 ecogroups”: a particularly complex polymorphic genomic block (reference genome 2, genome block 13, Ref2-13, 82 kb, Figure 3D, Supplementary file 1).

Figure 3D legend: …The numerical labels above reference genomes indicate the identity of the coadaptation genome blocks, corresponding to the information in Supplementary file 1.

Subsection “Marriage changes everything”. While the references to Henry VIII is funny, I'm not sure that all members of the international science community will understand.

We have altered the text to provide further context and make the text comprehensible to those who do not know who Henry VIII is.

Subsection “Marriage changes everything”: In this respect, the interaction between loci resembles the human institution of marriage, which has historically been thought of as an irreversible bond, although this notion loosened in the Christian world following the reign of King Henry VIII in England.

Subsection “Detection of coadapted loci”. A little more thorough description of SuperDCA and SpydrPick would be appreciated.

Thank you for pointing out this, we have described more details on these two tools as follows:

Subsection “Detection of coadapted loci”: SuperDCA is based on direct coupling analysis (DCA) model (17), a model to predict protein residue-residue contact from multiple sequence alignment, which has recently been extended to genome-wide epistasis analysis within bacterial populations. […] SpydrPick took one hour to finish the calculation using 32 cores and 1 Gb memory.

Figure 1. I don't think panel B is a Q-Q plot. It is really a frequency distribution of P-values. As mentioned for for subsection “Detection and characterization of interaction groups”, you haven't introduced EG1 yet, so minimally write out ecogroup.

Thank you for pointing out this. We moved this panel as Figure 2 and have revised the figure legend to clarify it.

“Figure 2. Frequency distribution of Fisher exact test *P* values between genetic variants. Colors and shapes indicate the interactions between different types of variants. The vertical dotted line shows the threshold *P* = 10^-10^.”

Figure 4. Although you state it in the legend, it would help readers if you labeled the x- and y-axes in panel A with 'Strains' and 'Loci'. The whole nomenclature is pretty confusing and is indecipherable to a reader who is skimming the figures. You should repeat some of the description you make in the text (around subsection “Detailed characterization of the largest interaction group”). Why is panel B broken out? Explain this in the legend. I still don't fully understand panel D. You state that 4 reference genomes are shown. Presumably these correspond to the text such as RIMD2210633, but this is not stated anywhere. Why do you have Ref1 and then 3 different Ref2s next to these designations? How come the tan and blue blocks on the left side of panel 1 don't have a designation? Are they all from Ref1: RIMD221063? I don't understand how that could be. And what exactly is the copying probability? This figure legend needs a lot of work.

We thank the reviewer for the constructive suggestions, we moved previous Figure 4 as Figure 3 and have extensively revised the figure and figure legend to make them clearer.

Figure 5. The legend is very non-descriptive and needs work.

We have moved previous Figure 5 as Figure 4 and revised the figure legend.

Bottom line regarding these figure legends. I was so annoyed trying to work through the figures that I almost suggested rejecting the manuscript. Fortunately, I decided to put it aside for the day and come back with a cooler head. But don't make this so hard on readers.

We are very sorry for not providing clear description on figures and figure legends. Thank you very much for giving us opportunity to revise. We have extensively revised the figure legends and hope this has now been rectified.

Reviewer 2:I would appreciate a more thorough description of the genome diversity used in different places in the study. Primarily, the interaction groups are identified using a very carefully curated set of strains without population structure, but analyses on the distribution of those SNPs and elements is made in a larger set that is not similarly characterized in the manuscript. I would like to see a similar figure for Figure 1A for the set of 469 strains used throughout the analysis.

Thank you this is an excellent suggestion, we now have a new Figure 1 that compares 469 strains with the 198, including PCoA. We believe that this new figure will help this reviewer and other readers to understand the pattern of diversity in the species much more easily. We have described different dataset at the beginning of Results section and performed PCoA analysis for two datasets (please see our response to the first comment).

Subsection “Detection and characterization of interaction groups”:

“The presence of clonal and population structure within a dataset results in genome-wide LD that can confound screens for epistatic interactions based on LD patterns. […] We then used fineSTRUCTURE (12) to look for signals of clonal structure within the remaining isolates, iteratively removing isolates until all of the isolates were assigned to a single population (Methods), leading to a discovery dataset of 198 strains (Figure 1C, Figure 1—figure supplement 1).”

Subsection “Complex structure of interaction group 1”: In order to interpret these associations in a broader ecological and geographical context, we performed further analyses on the interactions that we identified in the discovery dataset on the non-redundant set of 469 strains.

Additionally, the star phylogeny from Figure 1A convinces me that there is no population structure present in the dataset used for identifying the IGs, but I have a poor intuition for how closely related the strains are because I am unsure how their SNP filtering could affect the tree produced. I think a maximum likelihood ribosomal gene tree with the scale in substitutions per site would better orient readers to the scale of diversity included in these analyses. This would be especially interesting to know how the strain groups from IG1 heirarchical clustering maps onto this tree.

The new Figure 1 shows how IG1 groups map onto the tree (only EG1a is distinct). We did construct a ribosomal gene tree (Author response image 1), but in our view the main conclusion that can be drawn from it is that the 53 ribosomal genes provide substantially less information to infer relationships than the whole genome, for example providing a less clear split between populations. Since this is far from the point of our study, we have chosen not to include it. The new scale bars of NJ trees in Figure 1 shows the scale, which is substitutions per site and together with the new elements in Figure 1, we believe this gives a clear picture now of the overall pattern of diversity in the species.

**Author response image 1. respfig1:** Maximum likelihood tree of 469 non-redundant strains based on 53 concatenated ribosomal gene nucleotide sequences. Red branches indicate strains in the discovery dataset. The colored circles indicate populations (inner) and ecogroups (outer) according to the legend.

Related to this point of understanding the genome diversity of these strains as how they relate to the distribution of IGs, I would be curious to see if the strain groupings from the hierarchical clustering of IG1 could be reproduced by hierarchical clustering of strains based on flexible genome content.

Thank you for the suggestion and we are very happy to investigate this. We have now provided the hierarchical clustering results separately for accessory genes, as well as core genes as Figure 3—figure supplement 1A, B, and described the relationship in the text as follows.

Subsection “Complex structure of interaction group 1”: Similar but not identical structuring of strains was obtained if clustering was performed only using core genome SNPs (Figure 3—figure supplement 1A) or accessory genome elements (Figure 3—figure supplement 1B).

Regarding the analogy presented in Figure 6, it was not explained when it is introduced in the Discussion section and the figure legend for this figure is not detailed in its explanation of how to read the schematic so I am not confident in my understanding of the meaning of the various details included in it. I think the manuscript would benefit from a more explicit description of their model for how such complex interaction groups could come to exist in natural microbial populations.

We have revised the main text to further clarify the model, moved previous Figure 6 as Figure 8 and revised the figure and legends as follows:

Subsection “Most interactions between core and accessory genomes are casual”: “These selection pressures will tend to promote a genomic architecture that facilitates casual gain and loss of genetic elements from particular genomes (Figure 8a).”

Subsection “Sometimes it makes sense to go steady”: “This can lead to longer-lasting associations between genetic elements (Figure 8b).”

Subsection “Marriage changes everything”: “These arguments imply that the evolution of fastidiousness, like the evolution of promiscuity, can be self-reinforcing. At some point in the differentiation of EG1a strains from the remaining ones, the coadaptation of the core loci to each other is likely to have become more-or-less irreversible (Figure 8c).”

Subsection “Coadapted gene complexes as speciation triggers”: “As coadapted regions become more numerous, the proportion of recombination events between eco-groups that are maladaptive will increase, which might prompt the evolution of mechanistic barriers to genetic exchange between them (Figure 8d).”

Concerning making the results easier to follow, I recommend changing figure legends and sub-headings within the Results section to be more descriptive and convey the take-away message the reader should have from reading that figure or section. Similarly, summary and topic sentences throughout the results would make appreciating the implications of results easier.

Thank you for this suggestion, we have revised the legend and use new sub-heading in current version of manuscript. Hope now the description on results are clearer.

Subsection “Detailed characterization of the largest interaction group” – "more-or-less strongly associated" please clarify your wording here.

We have removed these words and further clarified the association as follows:

Subsection “Complex structure of interaction group 1”: Three of these groups, EG1a, EG1b and EG1d have a large number of variants (50-965) that are associated with them, and we used the clustering to sort these variants into tiers, with “Tier 1” (T1) corresponding to the variants that showed the strongest association for each EG (Figure 3, Table 2, Supplementary file 1).

Figure 4 legend – restate color scheme of heatmap in both Figure 3 and Figure 4 legends instead of referring to previous figure legend.

Thank you for this suggestion, we move previous Figure 3 and 4 as Figure 6 and 3 now and revised the legends to clarify these.